# TAS-GS: Integrating Topology, Appearance and Semantics for Sparse-View 3D Gaussian Splatting

## Abstract

We present *TAS-GS*, a framework that extends 3D Gaussian Splatting (3DGS) to sparse-view reconstruction by integrating topology, appearance and semantic priors. TAS-GS addresses key challenges of sparse-view 3DGS, including structural fragility, texture incoherency, and loss of fine details, through three modules: (i) a topology-aware graph regularizer that prunes floaters and bridges structural gaps, (ii) a GNN-based appearance propagation module that refines textures in weakly supervised regions, and (iii) a semantic-rarity and boundary-aware modulator that preserves fine details and underrepresented categories. All modules are applied only during training, and the final representation remains fully compatible with the standard 3DGS rasterizer. Extensive experiments on LLFF and Mip-NeRF 360 show that TAS-GS consistently outperforms state-of-the-art NeRF- and Gaussian-based methods across a wide range of sparsity levels. Ablation studies further confirm the effectiveness of each component in improving both quantitative metrics and perceptual quality. Our code is available at https://anonymous.4open.science/r/56165123.

## 1 Introduction

Novel view synthesis (NVS) (Avidan & Shashua, 1997; Watson et al., 2022) is a fundamental problem in computer vision with a wide range of applications, such as film and visual effects production, augmented reality, and robotics. High-quality NVS requires both accurate geometric reconstruction and faithful modeling of complex appearance, especially when extrapolating to unseen viewpoints.

Neural Radiance Fields (NeRF) (Mildenhall et al., 2020) established the paradigm of volumetric neural rendering, achieving impressive fidelity but requiring dense input views and incurring heavy computational cost. Subsequent extensions improved quality and scalability by addressing anti-aliasing and unbounded scenes (Barron et al., 2021; 2022; 2023), accelerating convergence with explicit encodings (Fridovich-Keil et al., 2022; Mueller et al., 2022), and enhancing rendering efficiency with voxel- or octree-based structures (Yu et al., 2021a; Fridovich-Keil et al., 2022). More recently, 3D Gaussian Splatting (3DGS) (Kerbl et al., 2023) advanced explicit neural scene representations by replacing costly volumetric ray marching with tile-based differentiable splatting, achieving state-of-the-art fidelity–efficiency trade-offs and enabling real-time photorealistic novel view synthesis.

Despite these advances, sparse-view reconstruction remains underexplored. With limited input supervision, optimization becomes ill-posed, often producing ambiguous geometry, floating artifacts, and blurred textures. NeRF-based methods address sparsity using priors such as depth supervision (Deng et al., 2021), unseen-view consistency (Niemeyer et al., 2022), frequency regularization (Yang et al., 2023), or semantic guidance (Jain et al., 2021), but they tend to converge slowly and yield unstable textures. Recent 3DGS variants mitigate floaters and structural collapse with structural priors and regularization. For example, FSGS (Zhu et al., 2024) introduces unpooling and virtual training views, SCGaussian (Cheng et al., 2024) enforces multi-view consistency across matched rays, and NexusGS (Zheng et al., 2025) incorporates epipolar geometry constraints for robust pruning of misplaced Gaussians. Other strategies include stochastic regularization (Park et al., 2025), cross-view co-regularization (Zhang et al., 2024), and depth-aware normalization (Li et al.,

Table 1: Comparison of sparse-view reconstruction methods. We classify the priors used by representative methods into three categories: **Topology** (global geometry/structure), **Appearance** (photometric/frequency cues), and **Semantics** (high-level guidance). ✓ denotes explicit use, ✗ denotes not considered. TAS-GS is the only method that integrates all three priors while maintaining real-time performance.

| Method | Views | Topology | Appearance | Semantics | Real-time | Remarks |
|---|---|---|---|---|---|---|
| NeRF (Mildenhall et al., 2020) | Dense | ✗ | ✓ | ✗ | ✗ | Implicit volumetric, high quality but slow |
| Mip-NeRF (Barron et al., 2021) | Dense | ✗ | ✓ | ✗ | ✗ | Cone sampling, anti-aliasing, robustness |
| IBRNet (Wang et al., 2021) | Sparse | ✗ | ✓ | ✗ | ✗ | Pixel-aligned interpolation for few views |
| DS-NeRF (Deng et al., 2021) | Sparse | ✓ | ✓ | ✗ | ✗ | Depth supervision for sparse-view stability |
| RegNeRF (Niemeyer et al., 2022) | Sparse | ✗ | ✓ | ✗ | ✗ | Unseen-view consistency regularization |
| GeoNeRF (Johari et al., 2022) | Sparse | ✓ | ✓ | ✗ | ✗ | Epipolar geometric priors |
| FreeNeRF (Yang et al., 2023) | Sparse | ✗ | ✓ | ✗ | ✗ | Frequency curriculum to reduce overfitting |
| SparseNeRF (Wang et al., 2023) | Sparse | ✗ | ✓ | ✗ | ✗ | Ranking-based depth distillation |
| 3DGS (Kerbl et al., 2023) | Dense | ✗ | ✓ | ✗ | ✓ | Explicit splats; real-time baseline |
| DNGaussian (Li et al., 2024) | Sparse | ✓ | ✓ | ✗ | ✓ | Global–local depth normalization |
| FSGS (Zhu et al., 2024) | Sparse | ✓ | ✓ | ✗ | ✓ | Gaussian unpooling, virtual-view consistency |
| CoR-GS (Zhang et al., 2024) | Sparse | ✓ | ✓ | ✗ | ✓ | Co-regularization to suppress floaters |
| MVSplat (Chen et al., 2024) | Sparse | ✗ | ✓ | ✗ | ✓ | Efficient multi-view Gaussian initialization |
| SCGaussian (Cheng et al., 2024) | Sparse | ✓ | ✓ | ✗ | ✓ | Matched-ray structure consistency |
| FreeSplat (Wang et al., 2024) | Sparse | ✗ | ✓ | ✗ | ✓ | Generalizable indoor free-view synthesis |
| DropGaussian (Park et al., 2025) | Sparse | ✓ | ✓ | ✗ | ✓ | Stochastic splat dropping for structure |
| NexusGS (Zheng et al., 2025) | Sparse | ✓ | ✓ | ✗ | ✓ | Epipolar depth priors, robust fusion/pruning |
| **TAS-GS (Ours)** | Sparse | ✓ | ✓ | ✓ | ✓ | Topology–Appearance–Semantics aware 3DGS |

2024), while generalizable splatting models (Wang et al., 2024; Chen et al., 2024) extend sparse-view training toward cross-scene generalization. However, most existing approaches focus on a single type of prior. This limitation motivates our work: a unified framework that integrates topology, appearance, and semantic cues for robust sparse-view reconstruction. A qualitative comparison of representative methods is given in Table 1.

We present TAS-GS, a topology–appearance–semantics aware framework for sparse-view 3DGS. Our design explicitly unifies three complementary forms of guidance. First, a topology-aware graph regularizer prunes floaters and repairs geometric voids, ensuring structural integrity before appearance learning. Second, a GNN-based appearance propagation module propagates color and opacity across geometry-aware neighborhoods with visibility-adaptive residual blending, improving photometric coherence under weak supervision. Third, a semantic-rarity and boundary-aware modulator emphasizes rare categories and occlusion edges, sharpening contours and preserving underrepresented structures. Our approach applies all modules exclusively during training, producing Gaussians that are fully compatible with standard 3DGS rasterization. In contrast, methods that rely on inference-time structures (Su et al., 2024; Feng et al., 2023) alter the Gaussian representation, thus limiting compatibility and increasing runtime complexity. Extensive experiments on LLFF (Mildenhall et al., 2019) and Mip-NeRF 360 (Barron et al., 2022) demonstrate that TAS-GS consistently outperforms NeRF- and Gaussian-based baselines, both quantitatively and qualitatively.

Our contributions are summarized as follows:

- We introduce TAS-GS, the first framework to jointly integrate topology-, appearance-, and semantics-aware priors for sparse-view 3D Gaussian Splatting.

- We design a topology-aware graph regularizer that enforces connectivity and removes floaters through graph-based pruning and hole repair.

- We propose a GNN-based appearance propagation module with visibility-adaptive residual blending to propagate reliable attributes into weakly supervised regions.

- We develop a semantic-rarity and boundary-aware modulator to focus supervision on rare classes and edge pixels, preserving fine structures and preventing texture bleeding.

## 2 RELATED WORK

**Novel View Synthesis.** NVS has become a key area in 3D vision and graphics for generating realistic renderings from multiple viewpoints. NeRF (Mildenhall et al., 2020) introduced implicit volumetric rendering, achieving high fidelity but relying on dense-view optimization. Later extensions

improved scalability: Mip-NeRF (Barron et al., 2021) and Mip-NeRF 360 (Barron et al., 2022) tackled anti-aliasing and unbounded scenes, while Zip-NeRF (Barron et al., 2023) introduced grid-based sampling. To accelerate training, explicit encodings like PlenOctrees (Yu et al., 2021a), Plenoxels (Fridovich-Keil et al., 2022), and Instant-NGP (Mueller et al., 2022) replaced MLPs. Methods like pixelNeRF (Yu et al., 2021b) and DietNeRF (Jain et al., 2021) focused on better generalization. Recent work, including 3DGS (Kerbl et al., 2023), enables real-time photorealistic rendering, inspiring variants for avatars and relighting. The field has evolved from implicit radiance fields (Mildenhall et al., 2020; Barron et al., 2021; 2022) to efficient explicit encodings (Yu et al., 2021a; Fridovich-Keil et al., 2022; Mueller et al., 2022), and real-time splatting (Kerbl et al., 2023), laying the groundwork for topology-, appearance-, and semantics-aware methods like TAS-GS.

**Sparse-View Reconstruction.** Sparse-view 3D reconstruction is ill-posed due to severe geometric ambiguity and under-constrained nature. To address this, recent methods leverage strong priors and regularization: DS-NeRF (Deng et al., 2021) introduces depth supervision, IBRNet (Wang et al., 2021) interpolates pixel-aligned features, and GeoNeRF (Johari et al., 2022) exploits epipolar geometry. Regularizers in RegNeRF (Niemeyer et al., 2022), FreeNeRF (Yang et al., 2023), and SparseNeRF (Wang et al., 2023) improve stability, while methods like DietNeRF (Jain et al., 2021) incorporate semantic cues for higher-level guidance. These advances highlight the need for integrating geometric, regularization, and semantic constraints—principles that inform our unified framework, TAS-GS, combining topology-, appearance-, and semantics-aware learning.

**Sparse-View 3D Gaussian Splatting.** Despite real-time rendering, 3DGS (Kerbl et al., 2023) suffers from floaters and geometric distortions under sparse views. Recent methods address these issues with structural priors: FSGS (Zhu et al., 2024) introduces unpooling and virtual-view consistency, SCGaussian (Cheng et al., 2024) enforces multi-view ray consistency, and NexusGS (Zheng et al., 2025) incorporates epipolar geometry constraints. Regularization strategies like DropGaussian (Park et al., 2025), CoR-GS (Zhang et al., 2024), and DNGaussian (Li et al., 2024) further improve stability. Generalizable models like FreeSplat (Wang et al., 2024) and MVSplat (Chen et al., 2024) enhance cross-scene training. TAS-GS builds on these advances by integrating topological, appearance, and semantic constraints for improved sparse-view reconstruction.

## 3 PRELIMINARIES: 3D GAUSSIAN SPLATTING

3DGS (Kerbl et al., 2023) represents a scene as a set of anisotropic Gaussians $\mathcal{S} = \{\mathcal{G}_i\}_{i=1}^N$, rendered through depth-ordered $\alpha$-compositing. Each Gaussian $\mathcal{G} = (\mathbf{p}, \mathbf{o}, \mathbf{s}, \mathbf{r}, \mathbf{k})$ is parameterized by a center $\mathbf{p} \in \mathbb{R}^3$, opacity $\mathbf{o} \in [0, 1]$, scale $\mathbf{s} \in \mathbb{R}^3$, quaternion rotation $\mathbf{r} \in \mathbb{R}^4$, and spherical harmonics (SH) coefficients $\mathbf{k} = \{\mathbf{k}^0, \ldots, \mathbf{k}^L\}$. The spatial covariance is defined as:

$$\mathbf{\Sigma} = R\,S\,S^\top R^\top,$$

where $S = \mathrm{diag}(\mathbf{s})$ and $R = \mathrm{quat2rot}(\mathbf{r})$. The opacity at a 3D position $\mathbf{x}$ is given by:

$$\alpha(\mathbf{x}, \mathcal{G}) = \mathbf{o} \cdot \exp\left(-\tfrac{1}{2}(\mathbf{x} - \mathbf{p})^\top \mathbf{\Sigma}^{-1}(\mathbf{x} - \mathbf{p})\right).$$

Appearance is expressed with SH functions along the viewing direction $\mathbf{d}$:

$$\mathbf{c}(\mathbf{d}, \mathcal{G}) = \mathbf{k}^0 + \sum_{l=1}^{L} H_l(\mathbf{d})\,\mathbf{k}^l,$$

where $H_l$ denotes the SH basis of degree $l$. During rasterization, each 3D Gaussian is projected to a 2D elliptical footprint on the image plane, whose size and orientation are determined by the Jacobian of the projection. The model is initialized with Gaussians from a Structure-from-Motion (SfM) reconstruction (Schonberger & Frahm, 2016). It is then optimized with a combination of an $\ell_1$ loss and a DSSIM loss, supplemented by optional depth alignment or opacity regularization terms.

## 4 THE TAS-GS FRAMEWORK

Sparse-view 3DGS often exhibits structural fragility, texture inconsistency, and loss of fine details due to limited supervision. We introduce TAS-GS, a framework that enhances 3DGS through explicit *topological*, *appearance*, and *semantic* priors. TAS-GS incorporates a topology-aware graph

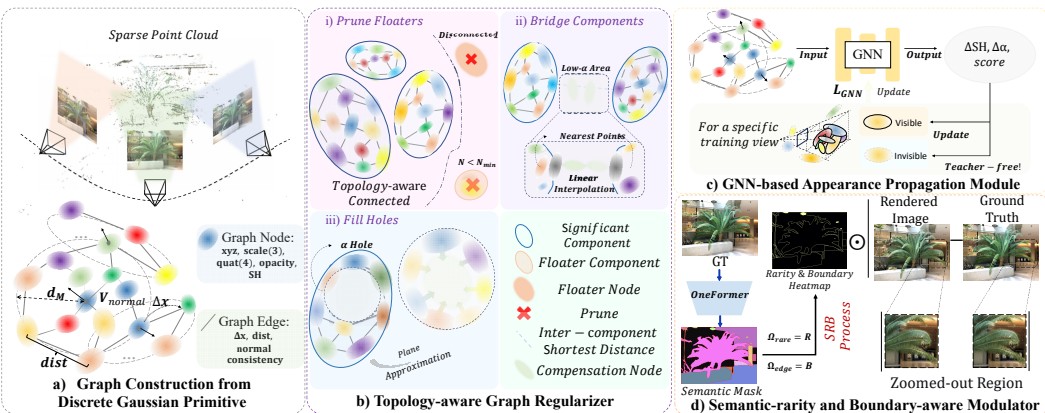

Figure 1: **Overview of TAS-GS.** (a) Gaussian Graph Construction: To address sparse inputs, we build a graph with geometric and appearance attributes for a unified representation. (b) Topology-aware Geometric Refinement: Under sparse supervision, we prune floaters, reconnect components, and fill holes to ensure coherent geometry. (c) Appearance Propagation: A GNN propagates appearance features across neighborhoods to resolve texture ambiguity after geometric stabilization. (d) Semantic Modulation: We apply modulation to underrepresented regions to sharpen contours and preserve fine details.

regularizer to eliminate floaters and repair holes, a GNN-based appearance propagation module for texture refinement in under-constrained areas, and a semantic modulator that emphasizes rare and boundary regions to recover fine details and underrepresented classes. All components support decoupled training and seamless integration, simultaneously ensuring real-time rendering and enhanced reconstruction quality under sparse-view conditions.

For clarity, we summarize key symbols and operators in the appendix (Tables 5 and 6). This includes Gaussian attributes (position, scale, rotation, opacity, spherical harmonics coefficients), graph construction operators, neighborhood relations, and loss formulations. Complexity-related symbols are detailed separately.

## 4.1 TOPOLOGY-AWARE GRAPH REGULARIZATION

Sparse-view reconstructions often suffer from disconnected components, floating Gaussians, and structural gaps. The objective of the topology-aware graph regularizer is to enforce geometric consistency by pruning spurious structures and repairing missing regions, thereby providing a stable foundation for subsequent appearance refinement.

We enrich the learnable Gaussian set $\mathcal{S} = \{\mathcal{G}_i\}_{i=1}^{N}$ with a topology-aware graph that supports pruning and hole repair before appearance refinement. Each Gaussian $\mathcal{G}_i$ has center $\mathbf{x}_i \in \mathbb{R}^3$ and covariance $\Sigma_i \in \mathbb{R}^{3\times3}$. We connect node $i$ to $k$ neighbors using a directional Mahalanobis metric:

$$d_M(i,j) = (\mathbf{x}_j - \mathbf{x}_i)^\top \Sigma_i^{-1}(\mathbf{x}_j - \mathbf{x}_i), \qquad \mathcal{N}_k(i) = \underset{j \neq i}{\arg\min}^{(k)} d_M(i,j). \tag{1}$$

The neighbor search is implemented in two stages. We first retrieve the top $K$ Euclidean candidates ($K = 100$ by default), and then refine them with equation 1 to select $k$ neighbors. Symmetrization yields an undirected adjacency $A$. Connected components $\{C_m\}_{m=1}^{M}$ are then computed, and small components are removed:

$$\mathcal{R} = \{i : i \in C_m, \ |C_m| < S_{\min}\}. \tag{2}$$

**Detecting geometric voids.** We score candidate viewpoints by the fraction of pixels with low opacity and choose:

$$c^\star = \underset{c}{\arg\max} \ \#\{u : \alpha_c(u) < \tau_\alpha\}, \qquad \tau_\alpha = 0.5. \tag{3}$$

We then render $\alpha_{c^\star}$ at native resolution and extract the 2D boundary with a $5 \times 5$ dilation:

$$\mathcal{B} = \mathcal{D}(\mathbf{1}[\alpha_{c^\star} < \tau_\alpha]) \setminus \mathbf{1}[\alpha_{c^\star} < \tau_\alpha]. \tag{4}$$

Boundary pixels $u \in \mathcal{B}$ are unprojected to 3D points $\mathbf{y}(u)$ using camera intrinsics and pose $(K, R, t)$. Each point is mapped to its nearest Gaussian with a KD-tree, giving a boundary set $\mathcal{I}_\partial = \{\mathrm{nn}(\mathbf{y}(u))\}$. Component labels $L(i) \in \{1, \ldots, M\}$ indicate whether a hole separates multiple components or lies within one.

**Inter-component bridging.** When $\mathcal{I}_\partial$ contains at least two distinct component labels, we connect the two most frequent ones, $\ell_1$ and $\ell_2$, by inserting $B$ Gaussians along their shortest cross-component segment. Specifically, we select candidate points:

$$\mathbf{p} \in \{\mathbf{x}_i : L(i) = \ell_1, i \in \mathcal{I}_\partial\}, \qquad \mathbf{q} \in \{\mathbf{x}_j : L(j) = \ell_2, j \in \mathcal{I}_\partial\},$$

and choose the closest pair $(\mathbf{p}^\star, \mathbf{q}^\star) = \arg\min_{\mathbf{p}, \mathbf{q}} \|\mathbf{p} - \mathbf{q}\|_2$. New Gaussians are then placed by linear interpolation:

$$\mathbf{x}_{\mathrm{new}}^{(b)} = (1 - t_b)\, \mathbf{p}^\star + t_b\, \mathbf{q}^\star, \quad t_b = \frac{b}{B+1}, \; b = 1, \ldots, B. \tag{5}$$

**Intra-component filling.** When only a single label $\ell$ is present, we fill the hole interior using local tangent-plane estimates. For a boundary subset $\mathcal{J} \subset \{i \in \mathcal{I}_\partial : L(i) = \ell\}$, each $i \in \mathcal{J}$ collects $k$ neighbors and computes the centroid $\boldsymbol{\mu}_i$ and covariance:

$$C_i = \frac{1}{k-1} \sum_{j \in \mathcal{N}_k(i)} (\mathbf{x}_j - \boldsymbol{\mu}_i)(\mathbf{x}_j - \boldsymbol{\mu}_i)^\top.$$

The plane normal $\mathbf{n}_i$ is taken as the eigenvector corresponding to the smallest eigenvalue of $C_i$. A new Gaussian is then initialized as:

$$\mathbf{x}_{\mathrm{new}}^{(i)} = \boldsymbol{\mu}_i, \qquad \mathbf{s}_{\mathrm{new}}^{(i)} = \left[0.5\bar{d}_i,\; 0.5\bar{d}_i,\; 0.1\bar{d}_i\right], \qquad \bar{d}_i = \frac{1}{k-1} \sum_{j \in \mathcal{N}_k(i)} \|\mathbf{x}_j - \boldsymbol{\mu}_i\|_2, \tag{6}$$

where the rotation is defined by aligning $\mathbf{n}_i$ with the $z$-axis and converted into a quaternion.

**Atomic execution and protection.** All edits are applied atomically at the end of each constraint stage. Gaussians referenced in equation 2 are pruned, while new Gaussians inherit spherical harmonics, opacity, scale, and rotation from their nearest neighbors, with opacity slightly reduced to ensure smooth integration. Inserted Gaussians are protected from pruning for $t_{\mathrm{grace}}$ iterations. This procedure runs periodically (every $T_{\mathrm{rasterize}}$ iterations) prior to the GNN stage and does not modify the rasterizer.

## 4.2 GNN-BASED APPEARANCE PROPAGATION

To compensate for weak supervision in sparse views, we propagate appearance across a geometry-aware graph and apply visibility-adaptive residual updates to the opacity and SH coefficients. Given $\mathcal{S} = \{\mathcal{G}_i\}_{i=1}^N$ with centers $\mathbf{x}_i$, covariances $\Sigma_i$, opacities $o_i$, and stacked SH vectors $\mathbf{k}_i$, we down-sample nodes $\mathcal{V}'$ with ratio $r$ and construct a $k$-nearest neighbor graph using the same procedure as in §4.1.

**Graph features.** Each node feature concatenates geometry and appearance:

$$\mathbf{h}_i^{(0)} = [\mathbf{x}_i, \mathbf{s}_i, \mathbf{q}_i, o_i, \mathrm{vec}(\mathbf{k}_i)], \quad \mathbf{e}_{ij} = [(\mathbf{x}_i - \mathbf{x}_j),\, \|\mathbf{x}_i - \mathbf{x}_j\|_2,\, \langle \mathbf{n}_i, \mathbf{n}_j \rangle],$$

where $\mathbf{n}_i$ is the shortest-axis normal estimated from $\Sigma_i$. Features and edges are standardized per batch, and scales are clamped for numerical stability.

**Message passing.** We adopt a three-layer edge-conditioned GATv2 (Brody et al., 2022). For layer $\ell$ and head $h$, the attention coefficient is defined as:

$$\alpha_{ij}^{(h)} = \mathrm{softmax}_{j \in \mathcal{N}(i)} \left( \mathbf{a}^{(h)\top} \sigma\big(W^{(h)}[\mathbf{h}_i^{(\ell)} \| \mathbf{h}_j^{(\ell)} \| U^{(h)} \mathbf{e}_{ij}]\big) \right), \tag{7}$$

and the node update is given by:

$$\mathbf{h}_i^{(\ell+1)} = \phi\left(\Big\|_{h=1}^{H} \sum_{j\in\mathcal{N}(i)} \alpha_{ij}^{(h)} V^{(h)}\mathbf{h}_j^{(\ell)}\right),$$

(8)

where $\phi(\cdot) = \tanh(\cdot)$, $\mathcal{N}(i)$ denotes the neighborhood of node $i$, and $H$ is the number of attention heads.

**Residual updates.** The MLP head predicts $\widehat{\boldsymbol{\delta}}_i = [\Delta o_i, \Delta\mathbf{k}_i, score_i]$ for $i \in \mathcal{V}'$. Only the appearance-related residuals are updated through visibility-adaptive blending:

$$\beta = w_{\text{res}}(1-\bar{v}), \qquad (o_i, \mathbf{k}_i) \leftarrow (o_i, \mathbf{k}_i) + \beta\,[\Delta o_i, \Delta\mathbf{k}_i], \quad \text{if } v_i = 0: \ \beta \leftarrow 2\beta,$$

(9)

where $\mathbf{v} \in \{0,1\}^N$ is the visibility mask and $\bar{v} = N^{-1}\sum_i v_i$.

**Survival score.** After temperature scaling, $score_i' = \sigma(score_i/\tau_s) \in (0,1)$ serves as a pruning prior. A Gaussian is pruned only if $o_i < o_{\min}$ and $score_i' < \tau_{\text{prune}}$, preventing premature deletion of structurally important but temporarily transparent splats.

**Visibility-aware reconstruction.** Let $\mathbf{a}_i = [o_i, \text{vec}(\mathbf{k}_i)]$ and $\widehat{\mathbf{a}}_i = \mathbf{a}_i + \widehat{\boldsymbol{\delta}}_i^{\text{app}}$. We apply a visibility-aware reconstruction loss on $\mathcal{V}'$:

$$\mathcal{L}_{\text{vis}} = \frac{1}{|\{i\in\mathcal{V}':v_i=1\}|} \sum_{i\in\mathcal{V}':v_i=1} \|\widehat{\mathbf{a}}_i - \mathbf{a}_i\|_2^2,$$

(10)

$$\mathcal{L}_{\text{inv}} = \frac{1}{|\{i\in\mathcal{V}':v_i=0\}|} \sum_{i\in\mathcal{V}':v_i=0} \|\mathbf{a}_i - \text{sg}(\widehat{\mathbf{a}}_i)\|_2^2,$$

(11)

$$\mathcal{L}_{\text{GNN}} = 0.1\,\mathcal{L}_{\text{vis}} + 1.0\,\mathcal{L}_{\text{inv}}.$$

(12)

The invisible loss is teacher-free: predictions serve as fixed targets, which prevents collapse and encourages propagation into occluded regions. $\mathcal{L}_{\text{GNN}}$ is combined with the renderer's photometric loss, and smoothness emerges naturally from message passing on the Mahalanobis graph without requiring an explicit Laplacian term.

## 4.3 SEMANTIC-RARITY AND BOUNDARY-AWARE MODULATION

We incorporate semantic priors to strengthen supervision on pixels that are (i) semantically rare or (ii) located at semantic boundaries, both of which are typically underconstrained in sparse views. Pseudo-labels are obtained from a frozen segmentation model (Jain et al., 2023), producing $S : \Omega \rightarrow \{0,\ldots,C-1\}$. Classes with pixel counts below the 25th percentile, excluding background, form the rare set $\mathcal{R}$. Boundaries are detected by changes in the four-neighborhood of labels:

$$\mathcal{B} = \{\mathbf{u} \in \Omega \mid S(\mathbf{u}) \neq S(\mathbf{u}+\mathbf{e}_x) \text{ or } S(\mathbf{u}) \neq S(\mathbf{u}+\mathbf{e}_y)\},$$

which emphasizes thin and occluding structures. Let $\hat{\mathbf{I}}$ denote the rendering and $\mathbf{I}$ the ground truth. We then define:

$$\Omega_{\text{rare}} = \{\mathbf{u} \in \Omega : S(\mathbf{u}) \in \mathcal{R}\}, \qquad \Omega_{\text{edge}} = \mathcal{B}.$$

The SRB loss is defined as:

$$\mathcal{L}_{\text{SRB}} = \lambda_{\text{rare}} \frac{1}{|\Omega_{\text{rare}}|} \sum_{\mathbf{u}\in\Omega_{\text{rare}}} \|\mathbf{I}(\mathbf{u}) - \hat{\mathbf{I}}(\mathbf{u})\|_1 + \lambda_{\text{edge}} \frac{1}{|\Omega_{\text{edge}}|} \sum_{\mathbf{u}\in\Omega_{\text{edge}}} \|\mathbf{I}(\mathbf{u}) - \hat{\mathbf{I}}(\mathbf{u})\|_1,$$

(13)

and can be equivalently written as a single weighted pass:

$$\mathcal{L}_{\text{SRB}} = \frac{1}{|\Omega|} \sum_{\mathbf{u}\in\Omega} w(\mathbf{u})\,\|\mathbf{I}(\mathbf{u}) - \hat{\mathbf{I}}(\mathbf{u})\|_1,$$

(14)

where

$$w(\mathbf{u}) = \begin{cases} w_{\text{rare}} & S(\mathbf{u}) \in \mathcal{R}, \\ w_{\text{edge}} & \mathbf{u} \in \mathcal{B}, \\ w_{\text{base}} & \text{otherwise}, \end{cases} \qquad w_{\text{rare}} = w_{\text{edge}} = 1.0, \ w_{\text{base}} = 0.2.$$

The loss is activated after a short warm-up and combined with other training objectives.

## 4.4 OPTIMIZATION

Our optimization jointly optimizes two aspects: (i) Gaussian attributes $\mathcal{G}$ and (ii) GNN parameters for appearance propagation. Densification and pruning follow 3DGS (Kerbl et al., 2023) but are guided by our graph propagation and semantic priors. This prevents uncontrolled growth while preserving rare structures, ensuring that the retained Gaussians remain both geometrically connected and semantically consistent.

The overall training objective is:

$$\mathcal{L} = \mathcal{L}_{\text{photo}} + \lambda_{\text{opacity}}\, \mathcal{L}_{\text{opacity}} + \lambda_{\text{app}}\, \mathcal{L}_{\text{GNN}} + \lambda_{\text{sem}}\, \mathcal{L}_{\text{SRB}}, \tag{15}$$

where $\mathcal{L}_{\text{photo}}$ combines $\ell_1$ and SSIM losses, and a entropy-style sparsity regularizer:

$$\mathcal{L}_{\text{opacity}} = -\frac{1}{|\mathcal{V}|} \sum_{i \in \mathcal{V}} \big[ o_i \log(o_i) + (1 - o_i) \log(1 - o_i) \big], \tag{16}$$

which encourages binary opacities for cleaner geometry. $\mathcal{L}_{\text{GNN}}$ supervises appearance propagation, and $\mathcal{L}_{\text{SRB}}$ emphasizes rare classes and boundary pixels. The weights $(\lambda_{\text{opacity}}, \lambda_{\text{app}}, \lambda_{\text{sem}})$ balance accuracy, stability, and compactness. Training runs for $T$ iterations with periodic graph rebuilding (every $T_{\text{build}}$) and delayed activation of semantic priors after warm-up.

## 5 EXPERIMENTS

**Implementation details.** We implement our method by extending NexusGS (Zheng et al., 2025), training for 30k iterations on a single NVIDIA RTX 3090 GPU. Our representation maintains full compatibility with the standard 3DGS (Kerbl et al., 2023). We conduct a comprehensive evaluation on the LLFF (Mildenhall et al., 2019) and Mip-NeRF360 (Barron et al., 2022) benchmarks, reporting performance across a range of input view counts. Further details are provided in the appendix.

Table 2: **Quantitative comparison on the LLFF dataset.** The top three results are color-coded (red: best, orange: second, yellow: third), our method's results are highlighted in bold.

| Method | 3-view | | | 6-view | | | 9-view | | |
|---|---|---|---|---|---|---|---|---|---|
| | PSNR↑ | SSIM↑ | LPIPS↓ | PSNR↑ | SSIM↑ | LPIPS↓ | PSNR↑ | SSIM↑ | LPIPS↓ |
| Mip-NeRF (Barron et al., 2021) | 16.11 | 0.401 | 0.460 | 22.91 | 0.756 | 0.213 | 24.88 | 0.826 | 0.170 |
| DietNeRF (Jain et al., 2021) | 14.94 | 0.370 | 0.496 | 21.75 | 0.717 | 0.248 | 24.28 | 0.801 | 0.183 |
| RegNeRF (Niemeyer et al., 2022) | 19.08 | 0.587 | 0.336 | 23.10 | 0.760 | 0.206 | 24.86 | 0.820 | 0.161 |
| FreeNeRF (Yang et al., 2023) | 19.63 | 0.612 | 0.308 | 23.73 | 0.779 | 0.195 | 25.13 | 0.827 | 0.160 |
| SparseNeRF (Wang et al., 2023) | 19.86 | 0.624 | 0.328 | - | - | - | - | - | - |
| 3DGS (Kerbl et al., 2023) | 19.22 | 0.649 | 0.229 | 23.80 | 0.814 | 0.125 | 25.44 | 0.860 | 0.096 |
| DNGaussian Li et al. (2024) | 19.12 | 0.591 | 0.294 | 22.18 | 0.755 | 0.198 | 23.17 | 0.788 | 0.180 |
| FSGS (Zhu et al., 2024) | 20.33 | 0.702 | 0.204 | 24.26 | 0.827 | 0.124 | 25.56 | 0.858 | 0.111 |
| SCGaussian (Cheng et al., 2024) | 20.41 | 0.712 | 0.212 | 23.41 | 0.801 | 0.164 | 24.75 | 0.855 | 0.126 |
| NexusGS (Zheng et al., 2025) | 21.03 | 0.737 | 0.178 | 23.71 | 0.801 | 0.139 | 24.52 | 0.820 | 0.131 |
| **TAS-GS (Ours)** | **21.52** | **0.760** | **0.161** | **24.62** | **0.837** | **0.111** | **25.54** | **0.861** | **0.096** |

**Results on the LLFF dataset.** Table 2 presents quantitative comparisons under sparse-view settings. NeRF-based methods (Jain et al., 2021; Niemeyer et al., 2022; Yang et al., 2023; Wang et al., 2023) perform poorly with only 3 views, yielding blurry reconstructions. While the seminal 3DGS (Kerbl et al., 2023) achieves strong results with dense inputs, it also degrades significantly under extreme sparsity (19.86 dB PSNR at 3 views), revealing susceptibility to sparse supervision. Compared to recent 3DGS variants (Li et al., 2024; Zhu et al., 2024; Cheng et al., 2024; Zheng et al., 2025), our method achieves the best or second-best results across all regimes. With only 3 views, it attains 21.52 dB PSNR and 0.161 LPIPS, outperforming NexusGS by +0.5 dB and −0.017 LPIPS, while maintaining state-of-the-art performance at 6 and 9 views, respectively. Qualitative results in Figure 2 show that NeRF-based methods produce blurry edges. Although 3DGS and FSGS improve sharpness, they suffer from floaters and fragmented geometry. SCGaussian better preserves structure but introduces noise, and NexusGS improves stability yet still fails under extreme sparsity. In contrast, our approach reconstructs sharp stone edges in fortress, preserves fine structures in horns, and accurately captures contours in complex scenes such as leaves and orchids.

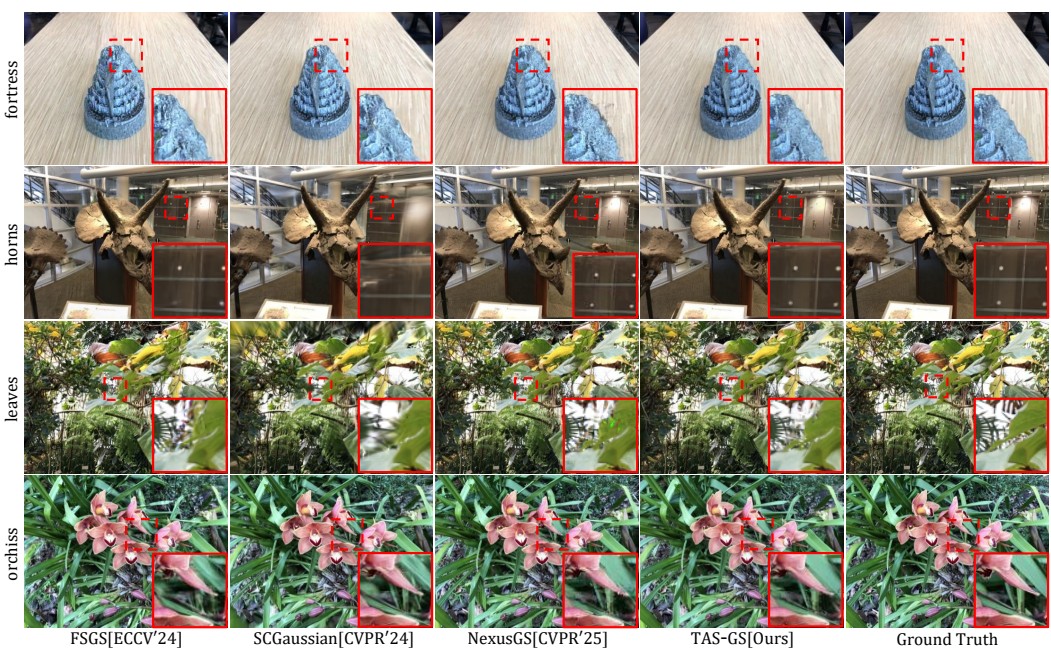

Figure 2: NVS results on the LLFF dataset (3 views).

Table 3: **Quantitative results on the Mip-NeRF360 dataset.** The top three results are color-coded (best: red; second: orange; third: yellow), and our results are highlighted in bold.

| Method | 12-view | | | 24-view | | |
|---|---|---|---|---|---|---|
| | PSNR↑ | SSIM↑ | LPIPS↓ | PSNR↑ | SSIM↑ | LPIPS↓ |
| Mip-NeRF360 (Barron et al., 2022) | - | - | - | 21.23 | 0.613 | 0.351 |
| DietNeRF (Jain et al., 2021) | - | - | - | 20.21 | 0.557 | 0.387 |
| RegNeRF (Niemeyer et al., 2022) | - | - | - | 22.19 | 0.643 | 0.355 |
| 3DGS (Kerbl et al., 2023) | 17.87 | 0.497 | 0.441 | 20.89 | 0.633 | 0.317 |
| FSGS (Zhu et al., 2024) | 17.94 | 0.510 | 0.445 | 21.70 | 0.674 | 0.323 |
| NexusGS (Zheng et al., 2025) | 18.59 | 0.499 | 0.376 | 22.40 | 0.687 | 0.237 |
| **TAS-GS (Ours)** | **19.25** | **0.539** | **0.364** | **22.92** | **0.716** | **0.226** |

**Results on the Mip-NeRF360 dataset.** We further evaluate our method on the Mip-NeRF360 dataset, with quantitative results summarized in Table 3. NeRF-based methods perform well with dense inputs, but degrade significantly under sparse settings, with the vanilla 3DGS dropping to 19.73 dB when only 3 views are provided. Compared to the 3DGS baselines, including DNGaussian (Li et al., 2024), FSGS (Zhu et al., 2024), SCGaussian (Cheng et al., 2024), and NexusGS (Zheng et al., 2025), our method achieves the best or second-best performance across all settings. Specifically, it reaches 20.81 dB PSNR and 0.198 LPIPS under the 3-view condition, surpassing NexusGS by +0.4 dB in PSNR and −0.012 LPIPS, while maintaining state-of-the-art results with 6 and 9 views. Qualitative comparisons are shown in Figure 3. NeRF-based methods suffer from blur and loss of detail; 3DGS (Kerbl et al., 2023) and FSGS(Zhu et al., 2024) improve sharpness but often introduce floaters and undesired thin structures. SCGaussian better preserves geometry but introduces texture noise, and NexusGS (Zheng et al., 2025) enhances stability yet still fails to recover fine details. In contrast, our approach reconstructs clean building edges in the garden scene, maintains thin structures such as poles and wires in the room, and produces sharp contours in the counter scene, resulting in more coherent and realistic textures.

**Ablation studies.** Figure 4 shows ablation results with components added progressively. The entropy-based opacity loss $\mathcal{L}_{\text{opacity}}$ raises PSNR/SSIM/LPIPS from 21.03/0.737/0.178 to 21.12/0.744/0.173, reducing floaters and allowing more pruning. Adding the topology-aware graph regularizer brings scores to 21.33/0.753/0.165 by stabilizing geometry and repairing thin

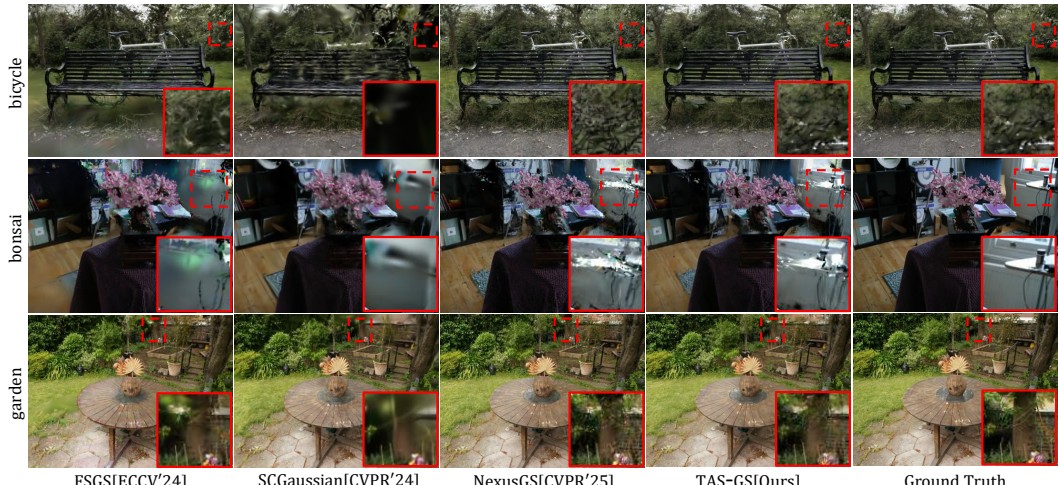

Figure 3: Qualitative comparison on the Mip-NeRF360 dataset (24 views).

| $\mathcal{L}_{\text{opacity}}$ | Topology | Appearance | Sematics | PSNR ↑ | SSIM ↑ | LPIPS ↓ |
|---|---|---|---|---|---|---|
| ✗ | ✗ | ✗ | ✗ | 21.03 | 0.737 | 0.178 |
| ✓ | ✗ | ✗ | ✗ | 21.12 | 0.744 | 0.173 |
| ✓ | ✓ | ✗ | ✗ | 21.33 | 0.753 | 0.165 |
| ✓ | ✓ | ✓ | ✗ | 21.41 | 0.758 | 0.163 |
| ✓ | ✓ | ✓ | ✓ | **21.52** | **0.760** | **0.161** |

GT   Baseline   $+L_{opacity}$   +Topology   +Appearance   +Sematics

Figure 4: Ablation study on the LLFF dataset with 3 views.

structures. The GNN-based appearance module further improves shading coherence, reaching 21.41/0.758/0.163 without inference cost. With the full model, we achieve the best results: 21.52 PSNR, 0.760 SSIM, and 0.161 LPIPS, preserving sharp contours and rare details. Visually, each component progressively enhances edge sharpness and texture quality. The final model remains compatible with the standard 3DGS rasterizer, maintaining real-time rendering while delivering superior reconstruction.

## 6 CONCLUSION

This paper presents **TAS-GS**, a topology–appearance–semantics-aware framework that extends 3D Gaussian Splatting for sparse-view novel view synthesis. In contrast to prior approaches, the proposed method effectively addresses key challenges including floaters, structural fragmentation, weak supervision, and underrepresented semantics. Extensive evaluations on LLFF and Mip-NeRF360 demonstrate consistent improvements over both NeRF- and Gaussian-based baselines in terms of distortion and perceptual metrics, while maintaining full compatibility with the standard 3DGS rasterizer. Ablation studies further validate the complementary benefits of each component, leading to cleaner textures, and more faithful reconstructions.

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

# A APPENDIX

## A.1 EXTENDED IMPLEMENTATION DETAILS

We provide the full set of hyperparameters and scheduling rules for reproducibility. Unless otherwise specified, all experiments are trained with Adam for 30k iterations on a single NVIDIA RTX 3090 GPU.

**Learning rates.** The Gaussian center learning rate decays exponentially from $1.6\times10^{-4}$ to $1.6\times10^{-6}$ with a delay multiplier of 0.01. Other attributes use fixed learning rates: $2.5\times10^{-3}$ for SH features, $5\times10^{-2}$ for opacity, $3\times10^{-2}$ for scaling, and $1\times10^{-3}$ for rotation.

Table 4: Training hyperparameters of our framework.

| Parameter | Value |
|---|---|
| Optimizer | Adam |
| Iterations | 30k |
| Learning rate (centers) | $1.6\times10^{-4} \to 1.6\times10^{-6}$ (exp. decay) |
| Learning rate (SH) | $2.5\times10^{-3}$ (fixed) |
| Learning rate (Opacity) | $5\times10^{-2}$ (fixed) |
| Learning rate (Scaling) | $3\times10^{-2}$ (fixed) |
| Learning rate (Rotation) | $1\times10^{-3}$ (fixed) |
| Warm-up length | 2k iterations |
| Densification start | 500 iterations |
| Pruning start | 500 iterations |
| Densification interval | 100 iterations |
| Opacity reset interval | 3k iterations |
| Pruning threshold | 0.005 |
| Graph rebuild frequency | $T_{\text{build}}$ iterations |
| GNN subsample ratio | 0.25 |
| Graph neighbors | $k{=}8$ (Mahalanobis) |
| SRB activation | after 20k iterations |
| Final representation | Standard 3DGS-compatible set |

**Topology-aware graph regularizer.** Activated every 1k iterations after a 2k warm-up. Small components with fewer than 100 Gaussians are pruned. Holes are either bridged across components ($B{=}5000$) or filled by tangent-plane initialization. New Gaussians inherit attributes from neighbors, start with reduced opacity, and are protected from pruning for 1k iterations.

**Densification and pruning.** Following 3DGS (Kerbl et al., 2023), cloning or splitting occurs every 100 steps, opacity resets every 3k steps, and pruning combines low-opacity thresholding, radius-based culling, and GNN survival scores.

**GNN-based appearance propagation module.** We use a 3-layer GATv2 (Brody et al., 2022) with hidden size 64 and heads $[4, 4, 1]$. Graphs are constructed with $k = 8$ neighbors and rebuilt periodically once enabled. Updates are blended with visibility-adaptive weights (initialized at $0.1$). GNN refinement starts after 29.7k steps in long runs.

**Semantic-rarity and boundary-aware modulator.** Pseudo-labels are obtained from One-Former (Jain et al., 2023). Rare classes are defined as those below the 25th percentile in pixel count (excluding background), and boundaries are identified by 4-neighborhood changes. SRB loss is applied after 20k iterations with weights $w_{\text{rare}} = w_{\text{edge}} = 1.0$ and $w_{\text{base}} = 0.2$.

**Training-only modules.** All auxiliary components (graph regularizer, GNN propagation module, SRB modulator) are applied only during training. After optimization, the model reduces to a standard Gaussian set $(\mathbf{p}, o, \mathbf{s}, \mathbf{r}, \mathbf{k})$ fully compatible with the original 3DGS rasterizer.

## A.2 NOTATION TABLE

Table 5: Summary of main notations used in this paper.

| Symbol | Type | Description |
|---|---|---|
| $\mathcal{S} = \{\mathcal{G}_i\}_{i=1}^N$ | Set | Learnable set of Gaussians / set of $N$ Gaussians |
| $\mathbf{p}_i \in \mathbb{R}^3$ | Vector | Center position of Gaussian $\mathcal{G}_i$ |
| $\mathbf{\Sigma}_i \in \mathbb{R}^{3\times 3}$ | Matrix | Covariance matrix (from scale and rotation) |
| $\mathbf{o}_i \in [0,1]$ | Scalar | Opacity of Gaussian $\mathcal{G}_i$ |
| $\mathbf{s}_i \in \mathbb{R}^3$ | Vector | Scale parameters of Gaussian $\mathcal{G}_i$ |
| $\mathbf{q}_i \in \mathbb{R}^4$ | Vector | Rotation quaternion of Gaussian $\mathcal{G}_i$ |
| $\mathbf{k}_i$ | Vector | SH coefficients |
| $\mathcal{N}_k(i)$ | Set | $k$ nearest neighbors of node $i$ (under $d_M$) |
| $d_M(i,j)$ | Scalar | Directional Mahalanobis distance between $i$ and $j$ |
| $A$ | Matrix | Adjacency matrix of symmetrized graph |
| $C_m$ | Set | $m$-th connected component |
| $\mathcal{R}_{\text{prune}}$ | Set | Indices of pruned small components |
| $\alpha_c(u)$ | Scalar | Alpha value at pixel $u$ in camera $c$ |
| $\text{nn}(\mathbf{y})$ | Operator | Nearest neighbor in $\mathcal{S}$ to 3D point $\mathbf{y}$ |
| $\mathcal{D}(\cdot)$ | Operator | Morphological dilation operator |
| $\mathbf{1}[\cdot]$ | Operator | Indicator function (1 if condition holds, else 0) |
| $\mathcal{A} \setminus \mathcal{B}$ | Operator | Set difference |
| $\#$ | Operator | Cardinality (number of elements in a set) |
| $\mathcal{I}_\partial$ | Set | Boundary Gaussian indices for a hole |
| $\mathcal{R}_{\text{rare}}$ | Set | Rare semantic class indices |
| $S_{\text{seg}} : \Omega \to \{0, \dots, C-1\}$ | Mapping | Semantic segmentation map over pixel domain $\Omega$ |
| $\mathcal{B}$ | Set | Semantic boundary pixels (4-neighborhood changes) |
| $\parallel$ | Operator | Concatenation of feature vectors |
| $\|\cdot\|_2$ | Operator | Euclidean $\ell_2$ norm |
| $\langle\cdot,\cdot\rangle$ | Operator | Inner product |
| $\text{softmax}_{j\in\mathcal{N}(i)}(\cdot)$ | Operator | Softmax over neighbors of $i$ |
| $\text{sg}(\cdot)$ | Operator | Stop-gradient operation |
| $\text{clip}(x; a, b)$ | Operator | Element-wise clipping of $x$ to $[a, b]$ |
| $\arg\min^{(k)}$ | Operator | Indices of $k$ smallest elements |

**Additional notation for complexity analysis.** In addition to the main notation above, Table 6 lists symbols specific to complexity and memory analysis, covering subsampling, graph size, GNN architecture, and training schedules.

## A.3 ALGORITHMS

In this section, we present the complete algorithmic details of our method. We first outline the overall training procedure, followed by the core modules: topology-aware graph regularizer, GNN-based appearance propagation module, and semantic-rarity and boundary-aware modulator. Each algorithm is provided in pseudocode form for clarity and reproducibility.

**Training Loop** We summarize the end-to-end training procedure of our method. The schedule integrates photometric reconstruction, topology-aware graph regularizer, GNN-based appearance propagation module, and semantic-rarity and boundary-aware modulator.

Table 6: Notation used in the complexity and memory analysis.

| Symbol | Type | Description |
|---|---|---|
| $N$ | Scalar | Number of Gaussians in the scene |
| $K$ | Scalar | Euclidean pre-neighbor count (before Mahalanobis refinement) |
| $k$ | Scalar | Mahalanobis $k$-NN degree (graph connectivity) |
| $r$ | Scalar | Downsampling ratio for GNN nodes ($N' = rN$) |
| $N'$ | Scalar | Number of Gaussians after downsampling |
| $|E|$ | Scalar | Number of edges in the full graph ($\approx Nk$) |
| $|E'|$ | Scalar | Number of edges in the downsampled graph ($\approx N'k$) |
| $H$ | Scalar | Number of GNN attention heads |
| $d$ | Scalar | Hidden width per GNN head |
| $L$ | Scalar | Number of GNN layers |
| $|\Omega|$ | Scalar | Number of pixels per rendered training view |
| $C$ | Scalar | Number of candidate views for hole detection |
| $B$ | Scalar | Number of Gaussians inserted for bridging |
| $|\mathcal{J}|$ | Scalar | Number of boundary seeds used for local plane filling |
| $T_{\text{regularize}}$ | Scalar | Interval for triggering topology-aware graph regularizer |
| $T_{\text{build}}$ | Scalar | Interval for rebuilding GNN graphs |

---

**Algorithm 1** TAS-GS Training Loop (schedules per §4.4)

---

**Input:** $\mathcal{S}$ initialized from SfM, schedules $T_{\text{regularize}}$, $T_{\text{build}}$, warm-ups, weights ($\lambda_{\text{opacity}}, \lambda_{\text{app}}, \lambda_{\text{sem}}$)

1: **for** $t = 1 \ldots T$ **do**
2:     Render minibatch views, compute $\mathcal{L}_{\text{photo}} = \ell_1 + \text{DSSIM}$
3:     $\mathcal{L}_{\text{opacity}} \leftarrow -\frac{1}{|\mathcal{V}|} \sum_{i \in \mathcal{V}} \left[ o_i \log o_i + (1 - o_i) \log(1 - o_i) \right]$
4:     **if** $t \geq$ warm-up for regularization and $t \bmod T_{\text{regularize}} = 0$ **then**
5:         Run Alg. 2, 3, 4, 5
6:     **end if**
7:     **if** $t \geq$ warm-up for GNN and $t \bmod T_{\text{build}} = 0$ **then**
8:         Run Alg. 6 to update $\{o_i, \mathbf{k}_i\}$ on a downsample
9:         $\mathcal{L}_{\text{GNN}} \leftarrow 0.1 \, \mathcal{L}_{\text{vis}} + 1.0 \, \mathcal{L}_{\text{inv}}$ (cf. Eq. (9)–(10))
10:    **else**
11:        $\mathcal{L}_{\text{GNN}} \leftarrow 0$
12:    **end if**
13:    **if** $t \geq$ warm-up for SRB **then**
14:        $\mathcal{L}_{\text{SRB}} \leftarrow$ Alg. 7
15:    **else**
16:        $\mathcal{L}_{\text{SRB}} \leftarrow 0$
17:    **end if**
18:    $\mathcal{L} \leftarrow \mathcal{L}_{\text{photo}} + \lambda_{\text{opacity}} \mathcal{L}_{\text{opacity}} + \lambda_{\text{app}} \mathcal{L}_{\text{GNN}} + \lambda_{\text{sem}} \mathcal{L}_{\text{SRB}}$
19:    Update $\mathcal{S}$ and module parameters by Adam, apply 3DGS densification/pruning policy
20: **end for**
21: **return** Final $\mathcal{S}$ (standard 3D Gaussians $(\mathbf{p}, o, \mathbf{s}, \mathbf{r}, \mathbf{k})$)

---

**Topology-Aware Graph Regularizer** We detail the topology-aware graph regularizer, which prunes small components, detects geometric voids, and repairs holes via bridging or filling strategies.

---

**Algorithm 2** Build Mahalanobis Graph & Prune Small Components (cf. §4.1)

---

**Input:** Gaussian set $\mathcal{S}$, $K$ (Euclidean pre-neighbors), $k$ (Mahalanobis neighbors), $S_{\min}$
**Output:** Undirected graph $A$, component labels $L(\cdot)$, prune set $\mathcal{R}$
1: **for** $i = 1 \ldots N$ **do**                                              $\triangleright$ Euclidean pre-selection
2:     $\mathcal{N}_K^{\text{Euc}}(i) \leftarrow$ top-$K$ nearest by $\|\mathbf{x}_j - \mathbf{x}_i\|_2$
3:     $\mathcal{N}_k(i) \leftarrow \arg\min_{j \in \mathcal{N}_K^{\text{Euc}}(i)}^{(k)} d_M(i,j)$
4: **end for**
5: $A \leftarrow$ symmetrize $\{(i,j) \mid j \in \mathcal{N}_k(i)\}$
6: $\{C_m\}_{m=1}^M \leftarrow$ ConnectedComponents$(A)$, assign $L(i) \in \{1, \ldots, M\}$
7: $\mathcal{R} \leftarrow \{i \mid |C_{L(i)}| < S_{\min}\}$                    $\triangleright$ prune small components
8: **return** $A, L(\cdot), \mathcal{R}$

---

**Algorithm 3** Hole Detection & Boundary Extraction (cf. Eq. (2)–(3))

---

**Input:** Candidate camera set $\mathcal{C}$ (interpolated), opacity threshold $\tau_\alpha$, dilation $\mathcal{D}$
**Output:** Best view $c^\star$, boundary index set $\mathcal{I}_\partial$
1: **for** each $c \in \mathcal{C}$ **do**
2:     Render alpha map $\alpha_c(u)$, compute $s_c = \#\{u \mid \alpha_c(u) < \tau_\alpha\}$
3: **end for**
4: $c^\star \leftarrow \arg\max_c s_c$
5: $\mathcal{M} \leftarrow \mathbf{1}[\alpha_{c^\star} < \tau_\alpha]$
6: $\mathcal{B} \leftarrow \mathcal{D}(\mathcal{M}) \setminus \mathcal{M}$                                  $\triangleright$ 2D hole boundary
7: $\mathcal{I}_\partial \leftarrow \{\text{nn}(\mathbf{y}(u)) \mid u \in \mathcal{B}\}$              $\triangleright$ $\mathbf{y}(u)$ is back-projected 3D point
8: **return** $c^\star, \mathcal{I}_\partial$

---

**Algorithm 4** Regularization Edits: Bridging and Filling (cf. Eq. (4)–(6))

---

**Input:** $A$, labels $L(\cdot)$, boundary indices $\mathcal{I}_\partial$, bridge count $B$, neighbor count $k$
**Output:** Insert list $Ins$ (new centers/scales/rotations)
1: $\mathcal{L}_\partial \leftarrow \{L(i) \mid i \in \mathcal{I}_\partial\}$, $Ins \leftarrow \varnothing$
2: **if** $|\text{unique}(\mathcal{L}_\partial)| \geq 2$ **then**                         $\triangleright$ Inter-component bridging
3:     $(\ell_1, \ell_2) \leftarrow$ two most frequent labels in $\mathcal{L}_\partial$
4:     $\mathcal{P} \leftarrow \{\mathbf{x}_i \mid i \in \mathcal{I}_\partial, L(i) = \ell_1\}$,   $\mathcal{Q} \leftarrow \{\mathbf{x}_j \mid j \in \mathcal{I}_\partial, L(j) = \ell_2\}$
5:     $(\mathbf{p}^\star, \mathbf{q}^\star) \leftarrow \arg\min_{\mathbf{p} \in \mathcal{P}, \mathbf{q} \in \mathcal{Q}} \|\mathbf{p} - \mathbf{q}\|_2$
6:     **for** $b = 1 \ldots B$ **do**
7:         $t_b \leftarrow b/(B+1)$, $\mathbf{x}_{\text{new}}^{(b)} \leftarrow (1 - t_b)\mathbf{p}^\star + t_b\mathbf{q}^\star$
8:         $Ins \leftarrow Ins \cup \{\mathbf{x}_{\text{new}}^{(b)}\}$
9:     **end for**
10: **else**                                  $\triangleright$ Intra-component filling via local tangent plane
11:     $\mathcal{J} \leftarrow$ downsample of $\{i \in \mathcal{I}_\partial\}$
12:     **for** each $i \in \mathcal{J}$ **do**
13:         $\mathcal{N}_k(i) \leftarrow$ Mahalanobis neighbors, $\boldsymbol{\mu}_i \leftarrow \frac{1}{k}\sum_{j \in \mathcal{N}_k(i)} \mathbf{x}_j$
14:         $C_i \leftarrow \frac{1}{k-1}\sum_{j \in \mathcal{N}_k(i)}(\mathbf{x}_j - \boldsymbol{\mu}_i)(\mathbf{x}_j - \boldsymbol{\mu}_i)^\top$
15:         $\mathbf{n}_i \leftarrow$ eigenvector of $C_i$ with the shortest eigenvalue
16:         $\bar{d}_i \leftarrow \frac{1}{k}\sum_{j \in \mathcal{N}_k(i)} \|\mathbf{x}_j - \boldsymbol{\mu}_i\|_2$
17:         Initialize candidate: center $\boldsymbol{\mu}_i$, scale $[0.5\bar{d}_i, 0.5\bar{d}_i, 0.1\bar{d}_i]$
18:         Set rotation aligning $\mathbf{n}_i$ to $\hat{\mathbf{z}}$, append to $Ins$
19:     **end for**
20: **end if**
21: **return** $Ins$

---

---

**Algorithm 5** Atomic Execution & Protection (one regularize cycle)

---

**Input:** Prune set $\mathcal{R}$, insert list $Ins$, grace steps $t_{\text{grace}}$
**Output:** Updated $\mathcal{S}$ and protection flags $\pi(\cdot)$
1: Remove $\{\mathcal{G}_i \mid i \in \mathcal{R}\}$ from $\mathcal{S}$
2: **for** each proposed $\mathbf{x}^{\text{new}}$ in $Ins$ **do**
3:      $j \leftarrow \text{nn}(\mathbf{x}^{\text{new}})$
4:      Inherit SH/scale/rotation from $\mathcal{G}_j$, set opacity to $\eta o_j$ with $\eta \in (0, 1)$
5:      Add new Gaussian to $\mathcal{S}$ and set protection $\pi = t_{\text{grace}}$
6: **end for**
7: **return** $\mathcal{S}, \pi(\cdot)$

---

**GNN-Based Appearance Propagation Module** We describe the appearance refinement module, where a graph neural network propagates information across Gaussians with visibility-adaptive residual blending.

---

**Algorithm 6** GNN Propagation with Visibility-Adaptive Residual Blending and Survival Score (cf. Eq. (7)–(10))

---

**Input:** Gaussian set $\mathcal{S}$, downsampling ratio $r$, neighbor count $k$, learnable residual weight $w_{\text{res}}$, temperature $\tau_s$, pruning threshold $\tau_{\text{prune}}$
**Output:** Updated opacities and SH coefficients for nodes in $\mathcal{V}'$, cached survival scores for pruning
1: $\mathcal{V}' \leftarrow$ uniform downsample of indices with ratio $r$
2: Build $k$-NN graph on $\mathcal{V}'$ via Euclidean pre-selection ($K$) followed by Mahalanobis refinement (Alg. 2)
3: Construct node features $\mathbf{h}_i^{(0)}$ and edge features $\mathbf{e}_{ij}$, normalize per batch
4: **for** $\ell = 0 \dots 2$ **do**                         ▷ 3 GATv2 layers
5:      **for** each head $h$ **do**
6:          $\alpha_{ij}^{(h)} \leftarrow \text{softmax}_{j \in \mathcal{N}(i)}\Big(\mathbf{a}^{(h)\top} \sigma(W^{(h)}[\mathbf{h}_i^{(\ell)} \| \mathbf{h}_j^{(\ell)} \| U^{(h)}\mathbf{e}_{ij}])\Big)$
7:      **end for**
8:      $\mathbf{h}_i^{(\ell+1)} \leftarrow \tanh\Big(\|_h \sum_{j \in \mathcal{N}(i)} \alpha_{ij}^{(h)} V^{(h)} \mathbf{h}_j^{(\ell)}\Big)$
9: **end for**
10: **Prediction:** Predict residuals and survival scores via the MLP head:

$$\widehat{\boldsymbol{\delta}}_i = [\Delta o_i, \ \Delta \mathbf{k}_i, \ score_i], \quad \forall i \in \mathcal{V}'$$

11: **Visibility-adaptive blending:** Compute visibility $v_i \in \{0, 1\}$, mean visibility $\bar{v} = \frac{1}{|\mathcal{V}'|} \sum_{i \in \mathcal{V}'} v_i$, and base blend factor $\beta \leftarrow w_{\text{res}}(1 - \bar{v})$
12: **for** each $i \in \mathcal{V}'$ **do**
13:      **if** $v_i = 0$ **then**
14:          $\beta_i \leftarrow 2\beta$
15:      **else**
16:          $\beta_i \leftarrow \beta$
17:      **end if**
18:      $(o_i, \mathbf{k}_i) \leftarrow (o_i, \mathbf{k}_i) + \beta_i [\Delta o_i, \Delta \mathbf{k}_i]$
19: **end for**
20: **Survival score post-processing:** Map raw $score_i$ to $[0, 1]$ via temperature-scaled sigmoid:

$$score_i' \leftarrow \sigma(score_i / \tau_s)$$

21: **Caching for pruning:** Scatter $score_i'$ back to the full set (non-sampled indices unchanged), updating the model cache $score_i'$
22: **Optional local decision:** Define prune prior $\pi_i \leftarrow 1 - score_i'$. Mark $i$ as *candidate-to-prune* if $(o_i < o_{\min})$ and $(score_i' < \tau_{\text{prune}})$      ▷ Final pruning is executed during the densify/prune schedule
23: **return** Updated $\{o_i, \mathbf{k}_i\}_{i \in \mathcal{V}'}$ and cached $score_i'$

---

**Semantic-rarity and Boundary-aware Modulator** We present the semantic rarity & boundary-aware modulator, which emphasizes rare categories and preserves fine structures near semantic boundaries.

---

**Algorithm 7** Semantic-rarity and Boundary-aware (SRB) Modulator (cf. Eq. (11)–(13))

---

**Input:** Rendered $\hat{\mathbf{I}}$, ground truth $\mathbf{I}$, pseudo labels $S$, percentile $p$, weights $w_{\text{rare}}, w_{\text{edge}}, w_{\text{base}}$
**Output:** $\mathcal{L}_{\text{SRB}}$
1: Compute per-class pixel counts on a subset, $\mathcal{R} \leftarrow$ classes below the $p$-th percentile (excluding background)
2: $\mathcal{B} \leftarrow \{\mathbf{u} \mid S(\mathbf{u}) \neq S(\mathbf{u} + \mathbf{e}_x) \text{ or } S(\mathbf{u}) \neq S(\mathbf{u} + \mathbf{e}_y)\}$
3: **for** each pixel $\mathbf{u} \in \Omega$ **do**
4: $\qquad w(\mathbf{u}) \leftarrow \begin{cases} w_{\text{rare}}, & S(\mathbf{u}) \in \mathcal{R} \\ w_{\text{edge}}, & \mathbf{u} \in \mathcal{B} \\ w_{\text{base}}, & \text{otherwise} \end{cases}$
5: **end for**
6: $\mathcal{L}_{\text{SRB}} \leftarrow \frac{1}{|\Omega|} \sum_{\mathbf{u} \in \Omega} w(\mathbf{u}) \|\mathbf{I}(\mathbf{u}) - \hat{\mathbf{I}}(\mathbf{u})\|_1$
7: **return** $\mathcal{L}_{\text{SRB}}$

---

### A.4 COMPUTATIONAL COMPLEXITY AND MEMORY

In this section, we provide a detailed complexity and memory analysis of our method, aligning each term with the algorithms in Appendix A.4. We denote by $N$ the number of Gaussians, $K \gg k$ the Euclidean pre-neighbor count and the Mahalanobis $k$-NN degree, $r \in (0, 1]$ the downsampling ratio for the GNN, $N' = rN$ the downsampled node count, $|E| \approx Nk$ the edge cardinality on the full graph (and $|E'| \approx N'k$ on the subsample), $H$ the number of attention heads, $d$ the hidden width, $L$ the number of GNN layers, $|\Omega|$ the number of pixels per rendered view, $C$ the number of candidate views for hole detection, $B$ the number of bridging Gaussians, and $|\mathcal{J}|$ the number of boundary seeds used for local plane filling. We also use $T_{\text{regularize}}$ and $T_{\text{build}}$ for constraint and graph-building periods, respectively.

**Topology-aware graph regularizer (Algs. 2–5).** **(i) Euclidean pre-selection.** Building a KD-tree takes $O(N \log N)$ time and $O(N)$ memory, querying top-$K$ neighbors for all nodes costs $O(N(\log N + K))$, yielding a candidate set per node. **(ii) Mahalanobis refinement.** For each node we evaluate $K$ quadratic forms with a fixed $3 \times 3$ metric, giving $O(NK)$ arithmetic plus $O(NK)$ selection to retrieve top-$k$ (or $O(NK \log k)$ with partial sort). We store at most $O(NK)$ candidate indices during refinement and finally keep $O(Nk)$ neighbors. **(iii) Symmetrization & connected components.** Forming an undirected adjacency with at most $|E| \approx Nk$ edges and computing connected components via BFS/Union-Find costs $O(N+|E|) = O(N+Nk)$ time and $O(N+|E|)$ memory. **(iv) Hole detection (Alg. 3).** Scoring $C$ candidate views by the low-opacity ratio and computing a $5 \times 5$ dilation per view cost $O(C|\Omega|)$. Unprojecting $M$ boundary pixels and mapping to nearest Gaussians via a KD-tree gives $O(M \log N)$ time and $O(N)$ extra memory for the tree. **(v) Bridging and filling (Alg. 4).** Let the boundary sets of the two dominant components have sizes $m_1, m_2$. A naive closest-pair search takes $O(m_1 m_2)$, while a KD-tree implementation reduces this to $O((m_1+m_2) \log N)$. Inserting $B$ interpolated Gaussians is $O(B)$. For intra-component filling, each of the $|\mathcal{J}|$ seeds gathers $k$ neighbors and fits a local plane by $3 \times 3$ eigendecomposition, which is $O(|\mathcal{J}|k)$ time and $O(|\mathcal{J}|)$ memory. **(vi) Atomic execution (Alg. 5).** Pruning $|\mathcal{R}|$ nodes and inserting $B$ nodes is linear in the edit sizes, nearest-neighbor inheritance uses a single KD-tree query per insertion, $O(B \log N)$.

**GNN-based appearance propagation module (Alg. 6).** **(i) Downsampled graph.** With ratio $r$, we operate on $N' = rN$ nodes and $|E'| \approx N'k$ edges. If neighbors are rebuilt on the downsample, the Euclidean pre-selection plus Mahalanobis refinement mirror the costs above with $N$ replaced by $N'$. **(ii) Message passing.** For a GATv2 layer with edge features, the dominant term is linear in edges and width: $O(|E'|Hd)$ for attention/logits and aggregation, $L$ layers give $O(L|E'|Hd)$. Backprop introduces a constant-factor overhead (typically $\times 2-3$) but does not change asymptotics. **(iii) Residual write-back & visibility-adaptive blending.** Updating opacity and SH on the down-

sample is $O(N' d_{\text{app}})$ with $d_{\text{app}}$ the appearance sub-vector size (opacity + SH). **(iv) Survival-score caching.** Writing the temperature-scaled scores $score_i'$ back to a global buffer is $O(N')$ time and $O(N)$ memory for one scalar per Gaussian.

**Semantic-rarity and boundary-aware modulator (Alg. 7).** Computing per-class pixel counts on a subset of frames, extracting 4-neighborhood boundaries (a discrete morphological gradient), and applying the weighted $\ell_1$ loss are all linear in pixels: $O(|\Omega|)$ time with $O(|\Omega|)$ temporary memory for masks. In practice, weights $w(\mathbf{u})$ are computed on the fly and do not require storing full-size buffers across steps.

**Amortized per-iteration cost.** Let $\mathcal{C}_{\text{photo}}$ be the photometric rendering loss cost, $\mathcal{C}_{\text{regularize}}$ the regularization cost (build graph, hole detection, edit, and $\mathcal{C}_{\text{gnn}} = O(L|E'|Hd) = O(L\,r\,N\,k\,H\,d)$ the GNN pass (on the subsampled graph). With regularization triggered every $T_{\text{regularize}}$ iterations and GNN building/execution every $T_{\text{build}}$, the amortized complexity per iteration is

$$\mathcal{C}_{\text{iter}} \;=\; \mathcal{C}_{\text{photo}} \;+\; \frac{\mathcal{C}_{\text{regularize}}}{T_{\text{regularize}}} \;+\; \frac{\mathcal{C}_{\text{build(down)}}}{T_{\text{build}}} \;+\; \mathbf{1}[\text{GNN active}]\,\mathcal{C}_{\text{gnn}},$$

where $\mathcal{C}_{\text{build(down)}}$ denotes the optional neighbor rebuild on the downsample. Using typical settings ($k{=}8$, $K{=}100$, $r{\in}[0.2, 0.4]$, $L{=}3$, $H{=}4$, $d{=}64$), the dominant training overhead beyond $\mathcal{C}_{\text{photo}}$ is linear in $N$, specifically $O(r\,k\,N)$ up to the constant $LHd$, all planning terms are also near-linear because $K, k$ are fixed small constants.

**Memory footprint.** **Regularizer.** Storing Euclidean candidates and Mahalanobis neighbors requires $O(NK)$ and $O(Nk)$ indices, respectively, the CSR adjacency for connected components uses $O(N+|E|)$ space. KD-trees for nearest-neighbor queries are $O(N)$. **GNN.** For the downsample, node activations plus edge activations per layer take $O(N'd + |E'|d)$, and thus $O(L(N'd + |E'|d))$ if kept for backprop, in practice, gradient checkpointing reduces the peak factor without changing asymptotics. Survival scores add one scalar per Gaussian ($O(N)$). **SRB.** Pixel-wise weights and masks can be streamed, peak memory is $O(|\Omega|)$ for the active minibatch.

**Runtime neutrality.** All auxiliary modules—topology-aware graph regularizer, GNN refinement, and SRB weighting—are *training-only*. After optimization, the representation is a standard Gaussian set, inference uses the unmodified 3DGS tile-based rasterizer with the same asymptotic cost as the baseline.

**Summary.** Under fixed local connectivity ($k$) and constant-width GNN ($H, d, L$), both the time and memory overheads of our auxiliary components scale *linearly* with $N$ (up to downsampling factor $r$), while pixel-space terms are linear in $|\Omega|$. This preserves scalability to large scenes and maintains baseline inference complexity.

## A.5 LIMITATIONS

TAS-GS has two main limitations. First, in scenarios with very low or no overlap, the topology repair heuristic may erroneously connect geometrically unrelated structures, introducing local artifacts (see Figure 5). Second, our method is mainly designed for small-scale, static scenes and does not generalize to dynamic scenes or large-scale unbounded scenes. These limitations motivate future research directions.

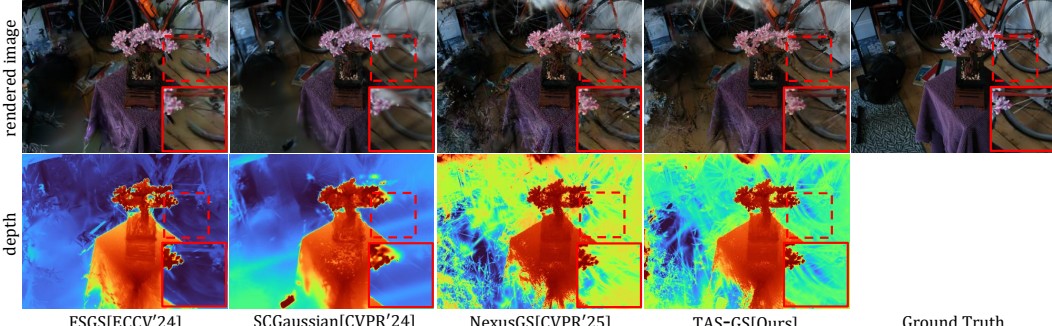

FSGS[ECCV'24]    SCGaussian[CVPR'24]    NexusGS[CVPR'25]    TAS-GS[Ours]    Ground Truth

Figure 5: **Qualitative comparisons on Mip-NeRF360 under low-overlap settings (12 Views).** Despite achieving more faithful rendering and smoother depth than competitors, our method can, in low-overlap scenarios, exhibit local artifacts when the topology repair erroneously connects distinct structures.

## A.6 THE USE OF LARGE LANGUAGE MODELS (LLMS)

In this paper, **LLMs are used solely to aid in refining and polishing the writing**. Their role is limited to improving sentence structure, clarity, and overall fluency. All content, ideas, and research are independently generated by the authors. The LLM-generated content is reviewed and integrated by the authors to ensure accuracy and coherence. LLMs are not involved in any data generation, experimentation, or technical methodology. The authors take full responsibility for the final content.

