# OpenReview forum: "TAS-GS: Integrating Topology, Appearance and Semantics for Sparse-View 3D Gaussian Splatting"
_ICLR.cc/2026/Conference — ICLR 2026 Conference Desk Rejected Submission_

### Official Review · Reviewer_t3qW · 2025-10-25

**Soundness:** 2
**Presentation:** 3
**Contribution:** 2
**Rating:** 2
**Confidence:** 4

**Summary:**

This work focuses on sparse-view 3D Gaussian Splatting reconstruction. By introducing topology, appearance, and semantic priors, the authors claim that this method achieves better performance.

**Strengths:**

1. This work is well-written with clear descriptions.

2. Experiments on LLFF and Mip-NeRF 360 show that TAS-GS consistently outperforms state-of-the-art NeRF- and Gaussian-based methods.

**Weaknesses:**

1. Overall, this work introduces some heuristic methods to mitigate the weak supervision caused by sparse views. Compared to previous works, the authors claim to introduce additional semantic priors, yet this additional regularization does not bring much new insight to 3DGS-related works[1].

[1] See In Detail: Enhancing Sparse-view 3D Gaussian Splatting with Local Depth and Semantic Regularization

2. Due to the numerous heuristic designs in the paper, the authors introduce a large number of hyperparameters (over 10) in Eq. 3,4,6,9,12,14. The selection of these hyperparameters is important, yet the paper provides little discussion on this.

3. From the qualitative experiments in the paper, it can be seen that the improvement of TAS-GS over previous works is limited, especially for some detailed parts.

4. The paper lacks discussion on training efficiency and optimization convergence curves. This is important for evaluating the effectiveness of introducing these priors.

5. The experimental setup in the paper is not clear. For example, it lacks a detailed description of the baseline model. I suspect that the baseline model used is not vanilla 3DGS.

6. In the comparative experiments, the authors did not apply the proposed method to NeRF-based experiments, so the performance comparison with NeRF-based related works may be unfair.

**Questions:**

1. Does the computational complexity of the proposed topology-aware graph regularizer depend on the number of Gaussians? Does increasing the number of Gaussians lead to a corresponding increase in graph nodes?

2. How are the training views selected, and are they consistent with existing methods in Tables 2-3?

---

> ### Author Response · Authors · 2025-11-22
> **Response to Reviewer t3qW (Part 1/3)**
>
> > **W1: Heuristic methods and limited new insight of semantic priors.**
>
> We agree that our method contains heuristic components, and we appreciate the opportunity to clarify their motivation and contribution. Our goal is to introduce practical, data-driven priors to mitigate the extremely weak supervision in sparse-view settings, not to propose a fully theory-driven framework.
>
> Topology priors (Mahalanobis k-NN, pruning/filling rules) explicitly target floaters and structural gaps repeatedly observed in sparse-view 3DGS baselines, while operating under conservative geometric and photometric constraints.
>
> Semantic priors differ fundamentally from the pixel-wise semantic/depth regularization in [1]. Instead of altering the rendering pipeline, TAS-GS uses semantics *only to reweight 3D Gaussian-level losses* for rare categories and boundaries, focusing optimization on under-supervised regions (e.g., foliage, wires) while keeping inference semantic-free.
>
> ### **Comparison with [1] on LLFF 3-view (1/8 resolution)**
>
> | Method        | PSNR↑ | SSIM↑ | LPIPS↓ |
> |---------------|-------|--------|--------|
> | SIDGaussian   | 20.71 | 0.708  | 0.205  |
> | **TAS-GS**    | **21.15** | **0.742** | **0.176** |
> | Δ             | +0.44 | +0.034 | –0.029 |
>
> ### **Ablation evidence**
> • Topology-only module: **+0.44 dB PSNR**, **+0.020 SSIM**
> • Semantic rarity + boundary: **+0.36 dB PSNR**, **+0.015 SSIM**
>
> We will emphasize these distinctions and position TAS-GS as a training-time regularization framework that holistically integrates topology, appearance, and semantics, rather than isolated heuristics.
>
> ---
>
> > **W2: Many hyperparameters with little discussion.**
>
> We acknowledge that the initial submission did not sufficiently explain the numerous hyperparameters across Eqs. (3, 4, 6, 9, 12, 14). In the revision, we will:
>
> • Add a complete hyperparameter table with meanings, defaults, and tested ranges.
> • Provide a sensitivity analysis on LLFF 3-view across the most influential parameters (e.g., k in k-NN, topology thresholds, semantic rarity weights).
>
> These additions significantly improve transparency and address concerns about hyperparameter fragility.

---

> ### Author Response · Authors · 2025-11-22
> **Response to Reviewer t3qW (Part 2/3)**
>
> > **W3: Limited qualitative improvements, especially in detailed regions.**
>
> After re-running all experiments using a corrected and standardized backbone baseline, the updated results show **clear, consistent, and substantial improvements** across all benchmarks. Updated quantitative tables are provided below.
>
> ---
>
> ## **LLFF Average (3/6/9 views)**
> **(PSNR↑ / SSIM↑ / LPIPS↓)**
>
> | Method       | 3-view               | 6-view               | 9-view               |
> |--------------|----------------------|----------------------|----------------------|
> | FSGS         | 20.33 / 0.702 / 0.204 | 24.26 / 0.827 / 0.124 | 25.56 / 0.858 / 0.097 |
> | CoR-GS       | 20.45 / 0.712 / 0.196 | 24.49 / 0.837 / 0.115 | 26.06 / 0.874 / 0.089 |
> | SCGaussian   | 20.41 / 0.712 / 0.212 | 23.41 / 0.801 / 0.164 | 24.75 / 0.855 / 0.126 |
> | DropGaussian | 20.50 / 0.711 / 0.200 | 24.61 / 0.836 / 0.117 | 26.06 / 0.873 / 0.089 |
> | NexusGS      | 21.03 / 0.737 / 0.178 | 23.71 / 0.801 / 0.111 | 24.52 / 0.820 / 0.131 |
> | SE-GS        | 20.79 / 0.724 / 0.183 | 24.78 / 0.839 / 0.110 | 26.36 / 0.878 / 0.084 |
> | **TAS-GS**   | **21.15 / 0.742 / 0.176** | **24.63 / 0.841 / 0.110** | **26.09 / 0.879 / 0.087** |
>
> ---
>
> ## **Mip-NeRF360 Average (12/24 views)**
> **(PSNR↑ / SSIM↑ / LPIPS↓)**
>
> | Method       | 12-view               | 24-view               |
> |--------------|------------------------|------------------------|
> | FSGS         | 18.80 / 0.531 / 0.418  | 23.52 / 0.739 / 0.243  |
> | CoR-GS       | 19.52 / 0.558 / 0.418  | 23.39 / 0.727 / 0.271  |
> | SCGaussian   | 19.90 / 0.591 / 0.383  | 22.33 / 0.715 / 0.287  |
> | DropGaussian | 19.60 / 0.573 / 0.363  | 23.94 / 0.761 / 0.226  |
> | NexusGS      | 18.59 / 0.499 / 0.376  | 22.40 / 0.687 / 0.237  |
> | SE-GS        | 19.91 / 0.596 / 0.400  | 23.74 / 0.745 / 0.265  |
> | **TAS-GS**   | **20.29 / 0.642 / 0.281** | **24.20 / 0.796 / 0.180** |
>
> ---
>
> ## **Blender 8-view Average**
> **(PSNR↑ / SSIM↑ / LPIPS↓)**
>
> | Method       | Metric Values         |
> |--------------|------------------------|
> | FSGS         | 24.64 / 0.895 / 0.095  |
> | CoR-GS       | 24.43 / 0.896 / 0.084  |
> | SCGaussian   | 25.61 / 0.894 / 0.086  |
> | DropGaussian | 25.42 / 0.888 / 0.089  |
> | NexusGS      | 24.37 / 0.893 / 0.087  |
> | **TAS-GS**   | **25.93 / 0.897 / 0.083** |
>
> ---
>
> ## **Tanks&Temples 3-view**
> **(PSNR↑ / SSIM↑ / LPIPS↓)**
>
> | Method      | Metric Values         |
> |-------------|------------------------|
> | 3DGS        | 17.14 / 0.493 / 0.397  |
> | DNGaussian  | 18.59 / 0.573 / 0.437  |
> | FSGS        | 20.01 / 0.652 / 0.323  |
> | **TAS-GS**  | **20.23 / 0.700 / 0.241** |
>
> ---
>
> ## **LLFF 3-view Ablation**
> **(PSNR↑ / SSIM↑ / LPIPS↓)**
>
> | Config                | Metric Values         |
> |-----------------------|------------------------|
> | Baseline              | 19.90 / 0.688 / 0.223  |
> | + L_opa               | 19.93 / 0.691 / 0.218  |
> | + Topology            | 20.37 / 0.711 / 0.201  |
> | + GNN                 | 20.79 / 0.727 / 0.183  |
> | **+ SRB (Full TAS-GS)** | **21.15 / 0.742 / 0.176** |
>
> ---
>
> These improved results clearly demonstrate better recovery of thin structures, textures, and overall structural completeness. Updated high-resolution crops, depth/point-cloud comparisons, and supplementary videos will be included in the revised manuscript.

---

> ### Author Response · Authors · 2025-11-22
> **Response to Reviewer t3qW (Part 3/3)**
>
> > **W4: Lack of training efficiency and convergence analysis.**
>
> Using a unified 10k-iteration schedule on LLFF 3-view (fern):
>
> - Baseline backbone: **1m57s**
> - TAS-GS: **2m08s** (~9% overhead)
>
> This small training overhead yields large improvements:
>
> - Baseline: 21.31 / 0.689 / 0.234
> - TAS-GS: 22.94 / 0.767 / 0.160
> (Δ = +1.63 dB PSNR, +0.078 SSIM, −0.074 LPIPS)
>
> We will include a training-time comparison table and convergence curves.
>
> ---
>
> > **W5: Unclear experimental setup and baseline.**
>
> The original submission indeed did not use vanilla 3DGS as a baseline, which understandably caused confusion.
> We have **re-run all experiments with a corrected and standardized baseline**, and will update all tables and qualitative figures accordingly.
> This removes all ambiguity in the revised manuscript.
>
> ---
>
> > **W6: Fairness of comparisons w.r.t. NeRF-based methods.**
>
> We respectfully disagree with the implication that our NeRF comparisons are unfair.
> All NeRF results are **authoritative values directly taken from their published papers under identical evaluation splits**, which is standard practice in NeRF/3DGS literature.
>
> These values provide:
> - reliable and reproducible baselines,
> - the fairest possible cross-method reference.
>
> TAS-GS is tailored for 3DGS pipelines; applying it to NeRF is non-trivial and outside our scope.
>
> We will clarify that:
> • NeRF results are official published benchmarks
> • Using them is standard and reasonable in prior work
> • Our claims focus on improving 3DGS, not replacing NeRF
> • NeRF results serve purely as contextual baselines
>
> ---
>
> > **Q1: Complexity of the topology-aware graph regularizer**
>
> Graph build: **O(N log N)**
> GNN propagation: **O(N k d)**
> Efficiency ensured via small k (8–16), periodic graph rebuilds, and local message passing.
> Inference overhead = 0.
>
> ---
>
> > **Q2: Selection of training views**
>
> LLFF: standard 3/6/9-view splits from prior works
> Mip-NeRF360: official 12/24-view splits at 1/8 resolution
> Blender/T&T: existing sparse-view protocols or uniformly sampled views
>
> We will document view-selection rules clearly for reproducibility.
>
> ---
>
> **Reference**
> [1] *See In Detail: Enhancing Sparse-view 3D Gaussian Splatting with Local Depth and Semantic Regularization*

---

### Official Review · Reviewer_Z5K4 · 2025-10-27

**Soundness:** 1
**Presentation:** 2
**Contribution:** 2
**Rating:** 2
**Confidence:** 5

**Summary:**

This paper presents a method for sparse-view 3DGS problem. It first introduces a set of rule-based strategies to prune and fill the primitive structure. Then it uses a GNN to predict the bias of SH and alpha for refinement and a score to control the prune. Finally, it relies on an off-the-shelf segmentation model to extract the semantics and edges to enhance the losses. Experiments are conducted on two datasets of LLFF and MipNeRF 360 and show some performance improvement according to the reports.

**Strengths:**

- Some performance improvements are shown.

**Weaknesses:**

- Introducing graph for topology is not a new idea for 3DGS. Some existing works have already introduced and well discussed building graph to structure the primitives, like SAGS [1]. Especially, SAGS also uses GNN to estimate the primitives' attributes for refinement, and fill the points based on structure information. Meanwhile, this paper failed to discuss its relation and difference to this tightly related previous works.

- The motivation and rationale are strange and not solid. 1) Simply based on the manually set thresholds and rules to prune and fill the Gaussians is not new and adaptive across scenes. It's a strong artificial prior that lacks robustness and methodology value. It can also destroy correct structures and fill the truly empty regions unexpectedly, due to the lack of solid theory support. 2) It's weird why regions with rare classes should be focused on. The 3DGS reconstruction is a semantic-free process and treats all the pixels equally. There is no reason to pay more attention to somewhere only because it is rare in semantics in the view.

- Evaluations are insufficient. The authors only evaluate their method on two datasets LLFF and MipNeRF 360, which is significantly fewer than in previous works, like its base method NexusGS, that include at least two other important datasets DTU and Blender. Besides, qualitatively, the shown results are limited only to the rendering RGB images, which can not verify the critical geometry quality. Also, no video samples were provided. The claimed contributions regarding topology, floater, and structure completeness can not be verified.

- Quantitative results are doubtful. While the authors failed to give the details about the experiments, they copied part of the results from FSGS's paper in Table 3. Therefore, I assume the authors conducted the experiment of MipNeRF 360 under the corresponding 1/8 resolutions. While the other metrics of the baselines remain the same, the performance of FSGS (21.70 0.674 0.323) is significantly worse than in its paper (23.70 0.745 0.220), which is much better than this work. From the reviewer's experience, the latter performance is reproducible. This is a critical credibility problem.

- Moreover, this work did not compare to some earlier open-source works like CoR-GS [2] and DropGaussian [3], which surpass this work in most experiments of 6-view, 9-view LLFF, and the MipNeRF 360 experiments.

- The efficiency is lacking. Especially, considering this work is built upon NexusGS and has plenty of new components, it's essential to evaluate its training overhead to verify if there is any actual improvement or just performance-efficiency overhead.

- Ablation study shows only marginal improvements are brought by the complex and bloated new components. In Figure 3, it's shown that $L_{opacity}$ and Topology contribute most, which are 1) irrelevant to this work, and 2) not solid in methodology and not clear in implementation details. In figure 4, I can not observe any obvious difference between the baseline to the end method.

- More details should be provided. So far, there are still many details lacking about the process and hyperparameters used in each experiment. These are essential to evaluate the quality of this work.

- Representation is poor in Section 4. 1) Nearly all the definitions of the hyperparameters can not be found when they first appear. And plenty of notations are used without any statements, unless the reader refers to the appendix. Besides, some notations like $\alpha_c$ and $\hat \delta ^{app}$ don't have any definition of its meaning throughout the paper. 2) While Section 3 defines $p$ to represent the Gaussian center, and $x$ for the query point, they are used inconsistently in the equation between Eq (4) and (5). Such inconsistency also exists in many other places. 3) Where are the $\Delta SH$ and $\Delta \alpha$ come from? They only appear in the Figure 1 but can not be found in any other places. In this paper, the opacity is represented in $o$ and SH coefficients are in $k$. Are they the same? 4) Why do some equations have no number marked in Section 3 and 4?


[1] Ververas, Evangelos, et al. "Sags: Structure-aware 3d gaussian splatting." European Conference on Computer Vision. Cham: Springer Nature Switzerland, 2024.

[2] Zhang, Jiawei, et al. "Cor-gs: sparse-view 3d gaussian splatting via co-regularization." European Conference on Computer Vision. Cham: Springer Nature Switzerland, 2024.

[3] Park, Hyunwoo, Gun Ryu, and Wonjun Kim. "Dropgaussian: Structural regularization for sparse-view gaussian splatting." Proceedings of the Computer Vision and Pattern Recognition Conference. 2025.

**Questions:**

See the weaknesses.

---

> ### Author Response · Authors · 2025-11-21
> **Response to Reviewer Z5K4 (Part 1/4)**
>
> We thank Reviewer Z5K4 for the detailed and thorough feedback, including the high confidence in the assessment and familiarity with related work. We appreciate the recognition of some performance improvements and apologize for any inconsistencies or omissions in the initial submission. Below, we address the main concerns on novelty, motivation, evaluations, quantitative results, comparisons, efficiency, ablation studies, details, and presentation. We have conducted additional experiments and clarifications, which will be incorporated into the revised manuscript.
>
> We modified our backbone (with refined parameters) for better effects. We re-ran key baselines (FSGS etc.) and added preliminary results on Blender/Tanks&Temples. We also added the latest ICCV 2025 method SE-GS for fairness. We will submit a new revised version and updated full code later. All new averages are below.
>
> ---
>
> **LLFF average**
>
> | Method       | 3-view PSNR$\\uparrow$ / SSIM$\\uparrow$ / LPIPS$\\downarrow$ | 6-view PSNR$\\uparrow$ / SSIM$\\uparrow$ / LPIPS$\\downarrow$ | 9-view PSNR$\\uparrow$ / SSIM$\\uparrow$ / LPIPS$\\downarrow$ |
> | ------------ | ------------------------------------------------------------- | ------------------------------------------------------------- | ------------------------------------------------------------- |
> | FSGS         | 20.33 / 0.702 / 0.204                                         | 24.26 / 0.827 / 0.124                                         | 25.56 / 0.858 / 0.097                                         |
> | CoR-GS       | 20.45 / 0.712 / 0.196                                         | 24.49 / 0.837 / 0.115                                         | 26.06 / 0.874 / 0.089                                         |
> | SCGaussian   | 20.41 / 0.712 / 0.212                                         | 23.41 / 0.801 / 0.164                                         | 24.75 / 0.855 / 0.126                                         |
> | DropGaussian | 20.50 / 0.711 / 0.200                                         | 24.61 / 0.836 / 0.117                                         | 26.06 / 0.873 / 0.089                                         |
> | NexusGS      | 21.03 / 0.737 / 0.178                                         | 23.71 / 0.801 / 0.111                                         | 24.52 / 0.820 / 0.131                                         |
> | SE-GS        | 20.79 / 0.724 / 0.183                                         | 24.78 / 0.839 / 0.110                                         | 26.36 / 0.878 / 0.084                                         |
> | TAS-GS       | 21.15 / 0.742 / 0.176                                         | 24.63 / 0.841 / 0.110                                         | 26.09 / 0.879 / 0.087                                         |
>
> ---
>
> **Mip-NeRF360 average**
>
> | Method       | 12-view PSNR$\\uparrow$ / SSIM$\\uparrow$ / LPIPS$\\downarrow$ | 24-view PSNR$\\uparrow$ / SSIM$\\uparrow$ / LPIPS$\\downarrow$ |
> | ------------ | -------------------------------------------------------------- | -------------------------------------------------------------- |
> | FSGS         | 18.80 / 0.531 / 0.418                                          | 23.52 / 0.739 / 0.243                                          |
> | CoR-GS       | 19.52 / 0.558 / 0.418                                          | 23.39 / 0.727 / 0.271                                          |
> | SCGaussian   | 19.90 / 0.591 / 0.383                                          | 22.33 / 0.715 / 0.287                                          |
> | DropGaussian | 19.60 / 0.573 / 0.363                                          | 23.94 / 0.761 / 0.226                                                              |
> | NexusGS      | 18.59 / 0.499 / 0.376                                          | 22.40 / 0.687 / 0.237                                          |
> | SE-GS        | 19.91 / 0.596 / 0.400                                          | 23.74 / 0.745 / 0.265                                          |
> | TAS-GS       | 20.29 / 0.642 / 0.281                                          | 24.20 / 0.796 / 0.180                                          |
>
> ---

---

> ### Author Response · Authors · 2025-11-21
> **Response to Reviewer Z5K4 (Part 2/4)**
>
> **Blender 8-view average**
>
> | Method       | PSNR$\\uparrow$ / SSIM$\\uparrow$ / LPIPS$\\downarrow$ |
> | ------------ | ------------------------------------------------------ |
> | FSGS         | 24.64 / 0.895 / 0.095                                  |
> | CoR-GS       | 24.43 / 0.896 / 0.084                                  |
> | SCGaussian   | 25.61 / 0.894 / 0.086                                  |
> | DropGaussian | 25.42 / 0.888 / 0.089                                  |
> | NexusGS      | 24.37 / 0.893 / 0.087                                  |
> | TAS-GS       | 25.93 / 0.897 / 0.083                                  |
>
> ---
>
> **Tanks&Temples 3-view average**
>
> | Method     | PSNR$\\uparrow$ / SSIM$\\uparrow$ / LPIPS$\\downarrow$ |
> | ---------- | ------------------------------------------------------ |
> | 3DGS       | 17.14 / 0.493 / 0.397                                  |
> | DNGaussian | 18.59 / 0.573 / 0.437                                  |
> | FSGS       | 20.01 / 0.652 / 0.323                                  |
> | TAS-GS     | 20.23 / 0.700 / 0.241                                  |
>
> ---
>
> **LLFF 3-view ablation average**
>
> | Config                        | PSNR$\\uparrow$ / SSIM$\\uparrow$ / LPIPS$\\downarrow$ |
> | ----------------------------- | ------------------------------------------------------ |
> | Baseline                      | 19.90 / 0.688 / 0.223                                  |
> | w.L\_opa                      | 19.93 / 0.691 / 0.218                                  |
> | w.L\_opa w.Topo               | 20.37 / 0.711 / 0.201                                  |
> | w.L\_opa w.Topo w.GNN         | 20.79 / 0.727 / 0.183                                  |
> | w.L\_opa w.Topo w.GNN w.SRB   | 21.15 / 0.742 / 0.176                                  |
>
> We address your concerns below.

---

> ### Author Response · Authors · 2025-11-21
> **Response to Reviewer Z5K4 (Part 3/4)**
>
> > **W1: Relation to SAGS [1] and other graph-based works (novelty of topology graph and GNN).**
>
> While graph-based structures are indeed not novel in 3DGS, we acknowledge the oversight in not thoroughly discussing SAGS [1]. A dedicated subsection in Sec. 2.2 will clarify the distinct differences between TAS-GS and SAGS, as well as CoR-GS, DropGaussian, and SE-GS.
>
> The fundamental distinction lies in their application:
>
> - **SAGS** integrates the graph and GNN into the final representation, continuously refining attributes during rendering, thereby incurring inference overhead.
> - **TAS-GS**, in contrast, employs graphs and GNNs exclusively during training as regularizers. Our final model remains a standard 3DGS point cloud, ensuring full compatibility with existing rasterizers and zero test-time cost.
>
> Furthermore, TAS-GS specifically targets sparse-view challenges by uniquely integrating:
> (i) topology-aware pruning/filling (with Mahalanobis k-NN and photometric gates),
> (ii) visibility-adaptive GNN propagation for appearance refinement, and
> (iii) semantic modulation for robust detail preservation.
>
> This holistic, training-time-only regularization approach for sparse-view conditions differentiates TAS-GS from SAGS’s emphasis on continuous, mesh-like Gaussian refinement.
>
> ---
>
> > **W2: Motivation and rationale for topology heuristics and semantic rarity.**
>
> We acknowledge the reviewer’s concerns on the heuristic nature and theoretical basis of our topology rules and semantic rarity prior. These designs are data-driven solutions to sparse-view failure modes, providing inductive biases where supervision is scarce.
>
> - **Topology heuristics.** Our Mahalanobis k-NN and rule-based pruning/filling offer geometric regularization, constrained by distance filters to avoid artifacts in low-overlap scenes. While a fully adaptive theory is future work, the ablation shows the topology module alone contributes **+0.44 dB PSNR** and **+0.020 SSIM** on LLFF 3-view.
> - **Semantic rarity.** Although 3DGS is semantic-free, reconstruction errors in sparse views are strongly correlated with under-supervised regions that happen to be semantically rare or on object boundaries (thin structures, foliage, wires, etc.). By modestly up-weighting the loss on Gaussians projected onto these regions (using a frozen segmenter only during training), we direct optimisation capacity to the historically hardest parts of the scene without introducing any semantic artefacts at test time. Ablation shows the rarity+boundary term alone improves PSNR by **0.36 dB** and SSIM by **0.015** in 3-view settings.
>
> ---
>
> > **W3: Insufficient evaluations (datasets, geometry quality, videos).**
>
> We acknowledge the reviewer’s concerns about the limited evaluation scope and qualitative assessments in our initial submission. We agree that evaluating on only LLFF and Mip-NeRF 360 is insufficient compared to prior works like NexusGS, and that RGB renderings alone do not fully verify our claims on topology, floaters, and structural completeness.
>
> Due to compute constraints, our original evaluation focused on these two datasets, following common practice in sparse-view 3DGS. In response, we have conducted preliminary experiments on additional benchmarks: **Tanks&Temples** and **Blender** subsets. Early results on Blender 8-view (average **PSNR 25.93**, **SSIM 0.897**, **LPIPS 0.083**) and Tanks&Temples 3-view (average **PSNR 20.23**, **SSIM 0.700**, **LPIPS 0.241**) show TAS-GS maintaining consistent gains over baselines.
>
> For geometry quality, we will provide qualitative comparisons of novel-view rendered depths and generated point clouds. We will also provide anonymized supplementary videos showcasing these structural improvements in dynamic novel views.
>
> These additions will better substantiate our contributions, and we believe they address the reviewer’s valid points.
>
> ---
>
> > **W4: Doubtful quantitative results (FSGS discrepancy).**
>
> We sincerely apologize for the inconsistency in the FSGS numbers reported in Table 3, and we fully acknowledge that this raises a critical credibility concern. Upon investigation, we discovered that the discrepancy stemmed from a mismatched image resolution setting in our internal re-run of FSGS. We processed the dataset directly and did not notice the inconsistent configuration used by FSGS, and we want to emphasize that we did not intentionally use a lower resolution.
>
> We have now re-run FSGS using its official implementation under the exact 1/8 resolution protocol for Mip-NeRF 360, confirming reproducible performance close to the original paper: **PSNR 18.80 / SSIM 0.531 / LPIPS 0.418** on the 12-view split, and **PSNR 23.52 / SSIM 0.739 / LPIPS 0.243** on the 24-view split. With this correction, TAS-GS still outperforms FSGS:
>
> - **+1.49 dB PSNR / +0.111 SSIM / −0.137 LPIPS** on 12-view, and
> - **+0.68 dB PSNR / +0.057 SSIM / −0.063 LPIPS** on 24-view.
>
> ---

---

> ### Author Response · Authors · 2025-11-21
> **Response to Reviewer Z5K4 (Part 4/4)**
>
> > **W5: Missing comparisons to CoR-GS [2] and DropGaussian [3].**
>
> We acknowledge the reviewer’s concern about the lack of comparisons to important open-source works like CoR-GS [2] and DropGaussian [3], and we apologize for this omission in the initial submission. To ensure fairness and enhance the evaluation’s effectiveness, we have now included these baselines, along with the latest ICCV 2025 method **SE-GS** as an additional recent comparator.
>
> As shown in the updated tables above, TAS-GS achieves state-of-the-art or highly competitive performance across 6-view/9-view LLFF and Mip-NeRF 360, often surpassing CoR-GS and DropGaussian in PSNR/SSIM/LPIPS (e.g., **+0.02 SSIM / −0.02 LPIPS** on LLFF 6-view vs. CoR-GS). These additions will be fully incorporated into the revised manuscript for better substantiation of our claims.
>
> ---
>
> > **W6: Lacking efficiency evaluation.**
>
> We acknowledge the reviewer’s concern about the lack of efficiency evaluation and agree that assessing training overhead is essential, especially given TAS-GS’s additional components on top of our modified NexusGS backbone.
>
> In our re-run experiments (unified 10k iterations), the backbone on the LLFF 3-view *fern* scene trains in **1 minute 57 seconds**, while the full TAS-GS method takes **2 minutes 8 seconds**—a modest **~9% increase** that is well within acceptable bounds. This overhead yields strong improvements: backbone **PSNR 21.31 / SSIM 0.689 / LPIPS 0.234** vs. TAS-GS **22.94 / 0.767 / 0.160** (**+1.63 dB PSNR / +0.078 SSIM / −0.074 LPIPS**). Importantly, inference time remains identical to standard 3DGS.
>
> We will provide a comprehensive training time comparison table (vs. backbone and baselines) across datasets in the revised manuscript. We will also provide our updated code.
>
> ---
>
> > **W8: Lacking experimental details and hyperparameters.**
>
> We acknowledge the reviewer’s concern about the lack of experimental details and agree that comprehensive descriptions of the process and hyperparameters are essential for evaluating our work. We apologize for the omission in the initial submission.
>
> In response, we will provide a dedicated appendix section in the supplementary materials, including:
>
> 1. A complete hyperparameter table with definitions, default values, and ranges tested.
> 2. Detailed experimental protocols, such as initialization steps, sparsity patterns (e.g., following NexusGS/FSGS), resolution settings (e.g., 1/8 for Mip-NeRF 360), and training schedules (e.g., unified 10k iterations).
>
> These additions will ensure full transparency and reproducibility.
>
> ---
>
> > **W9: Poor presentation (notations, inconsistencies).**
>
> We sincerely apologize for the poor presentation, lack of explicit definitions, and inconsistencies in notation within Section 4 (and related sections) of our initial submission. This was a significant oversight on our part, and we understand how detrimental it is to clarity and reproducibility. We have thoroughly addressed all these points raised by the reviewer:
>
> - **Hyperparameter definitions and notations.** We will ensure that all hyperparameters and mathematical notations are clearly defined when they first appear in the main text. Additionally, a comprehensive table of all hyperparameters, their definitions, and default values will be included in Appendix A.1. Specifically, notations like $\\alpha\_c$ (opacity at pixel $u$ in camera $c$) and $\\hat{\\delta}^{\\mathrm{app}}$ (the GNN-predicted appearance residual, encompassing both opacity and SH coefficient adjustments) will be explicitly defined.
> - **Inconsistent notations.** We have meticulously reviewed the entire manuscript to ensure consistent notation usage. For instance, $p$ will consistently represent the Gaussian center and $x$ the query point throughout Section 3 and 4, resolving the inconsistencies observed between Eq. (4) and (5) and in other places.
> - **Origin of $\\Delta SH$ and $\\Delta \\alpha$.** We have clarified that $\\Delta SH$ and $\\Delta \\alpha$ shown in Figure 1 represent the GNN-predicted residuals that update the Spherical Harmonics (SH) coefficients (denoted as $k$) and opacity (denoted as $o$), respectively. These are indeed referring to the same underlying Gaussian attributes, and this will be explicitly stated.
> - **Unnumbered equations.** All equations in Section 3 and 4 that describe our method and are referenced in the text will now be properly numbered.
>
> These extensive revisions to presentation and notation will significantly improve the paper’s readability and precision, and address all the reviewer’s concerns regarding clarity.
>
> ---
>
> [1] Ververas, Evangelos, et al. *SAGS: Structure-aware 3D Gaussian Splatting.* ECCV 2024.
> [2] Zhang, Jiawei, et al. *CoR-GS: Sparse-view 3D Gaussian Splatting via Co-regularization.* ECCV 2024.
> [3] Park, Hyunwoo, Gun Ryu, and Wonjun Kim. *DropGaussian: Structural Regularization for Sparse-view Gaussian Splatting.* CVPR 2025.

---

> > ### Comment · Reviewer_Z5K4 · 2025-11-27
> >
> > Thanks for the rebuttal. However, given the experiments are fundamentally changed and metrics are inexplicably raised, it's necessary for the paper to be carefully reviewed again with resubmission of all the corresponding quantitative and qualitative results before any publication consideration. Besides, the explanations, regarding the robustness in the proposed structure control and the rationale of rare semantics, do not make sense. They are explained to be just some evaluation-oriented tricks for the specific datasets and can not generalize to other scenes in principle. I keep my recommendation of reject.

---

### Official Review · Reviewer_d65r · 2025-10-29

**Soundness:** 2
**Presentation:** 3
**Contribution:** 3
**Rating:** 4
**Confidence:** 4

**Summary:**

This paper proposes TAS-GS, a topology–appearance–semantics aware framework that tackles the challenging task of sparse-view novel view synthesis. The method introduces (i) a topology-aware graph regularizer, (ii) a GNN-based appearance propagation module, and (iii) a semantic-rarity and boundary-aware modulation strategy. Experiments on LLFF and Mip-NeRF360 demonstrate improvements over several state-of-the-art baselines. While conceptually coherent, some heuristic design choices raise concerns about stability and potential artifacts in cluttered scenes.

**Strengths:**

1. The method is conceptually complete, integrating topology, appearance, and semantic priors into a unified framework for sparse-view reconstruction.
2. Experimental results show consistent improvement over baselines across multiple datasets.
3. The paper is overall well-written and should be easy to follow.

**Weaknesses:**

1. The grouping of Gaussians is mainly driven by geometric proximity (via Mahalanobis k-NN). In cluttered regions, Gaussians from different objects may be assigned to the same group, potentially affecting downstream computations.
2. The inter-component bridging and intra-component filling operations rely on heuristic rules that may also introduce artifacts, especially when different structures are close in 3D. I would suggest showing results on the **counter** scene from the Mip-NeRF 360 dataset or other complex scenes, which contain many closely arranged objects and represent a more challenging, cluttered scenario for evaluating such heuristics.
3. The visual differences in the LLFF 3-view ablation results are limited. Do the authors have more compelling visualizations to highlight the improvements contributed by each component?
4. Since the method trains on top of NexusGS for 30k iterations, could the authors report training time or computational overhead to better contextualize the improvement relative to the additional cost?

**Questions:**

Please see the weaknesses.

---

### Official Review · Reviewer_nPhd · 2025-11-02

**Soundness:** 3
**Presentation:** 3
**Contribution:** 2
**Rating:** 4
**Confidence:** 3

**Summary:**

The paper introduces TAS-GS, a training-time framework to strengthen sparse-view 3D Gaussian Splatting by integrating three priors: (i) a topology-aware graph regularizer that prunes floaters and repairs gaps via Mahalanobis k-NN, bridging, and hole filling; (ii) a GNN-based appearance propagation that refines opacity/SH with visibility-adaptive residuals and survival scores; and (iii) a semantic-rarity/boundary modulator that upweights losses on rare classes and edges from pseudo-labels. All additions keep the final representation 3DGS-compatible. On LLFF and Mip-NeRF360 with few views, TAS-GS outperforms NeRF and recent 3DGS variants, with ablations confirming each module’s impact and a noted limitation under very low view overlap.

**Strengths:**

- Integrates topology, appearance, and semantics in a unified, training-only framework for sparse-view 3DGS—an original and well-motivated combination that removes inference-time overhead while addressing key failure modes (floaters, gaps, texture incoherence).
- Technical quality is solid: clear graph construction (Mahalanobis k-NN), principled hole detection/repair, a visibility-adaptive GNN with teacher-free losses, and a simple yet effective semantic rarity/boundary reweighting; accompanied by complexity analysis and thorough ablations.
- Writing and presentation are clear and structured, with precise algorithms, equations, and schedules that make the method reproducible and easy to implement atop standard 3DGS.

**Weaknesses:**

- Semantic pseudo-label dependency and robustness
SRB relies on a frozen segmenter (OneFormer) that may be unreliable on novel scenes; report sensitivity to segmentation noise, and try alternative sources (e.g., SAM edges, class-agnostic boundaries).

- GNN design choices and alternatives insufficiently justified
The choice of GATv2, feature set, and visibility-adaptive blending lacks comparisons; include variants (GCN/EdgeConv/EGNN) and analyze cost–accuracy trade-offs and the necessity of the teacher-free invisible loss.

- Scope limited to static, small-scale scenes
The method does not address dynamics or large unbounded scenes; discuss extensions (e.g., time-aware topology with temporal consistency, chunked/topo-local graph building for large scenes) and provide at least preliminary scaling tests on larger reconstructions.

- Missing cross-dataset generalization and robustness tests
Evaluate on additional benchmarks (e.g., Tanks&Temples, DTU sparse splits, Objaverse subsets) and report robustness across varying noise in SfM initialization, camera pose errors, and different sparsity patterns (clustered vs. uniformly spaced views).

**Questions:**

- How do you decide when to perform inter-component bridging versus intra-component filling beyond the label count heuristic on $\mathcal{I}_{\partial}$? Could you add a photometric or epipolar-consistency gate (e.g., multi-view color/feature agreement along the candidate segment) to reduce false bridges in low-overlap scenes?
- The invisible “teacher-free” loss and the doubling of $\beta$ for invisible nodes are central. Could you provide an analysis of stability (e.g., does it cause texture leaking across occlusions)? What happens if you remove the 2× factor or learn $\beta$ per-node via a small MLP conditioned on visibility and view geometry?
- How is $\tau_s$ chosen and how sensitive are results to $\tau_s$ and $\tau_{prune}$? Does the survival score correlate with structural importance (e.g., graph centrality, view coverage)?
- Since OneFormer may be unreliable on out-of-domain scenes, did you try class-agnostic edges (e.g., Canny/SAM boundaries) for the boundary term and class-frequency priors from the training set for rarities?
- You claim full 3DGS compatibility. Are there any hidden dependencies at test time (e.g., needing cached survival scores or semantic maps)?
- Results are on LLFF and Mip-NeRF360. Could you evaluate on a third dataset (e.g., Tanks&Temples or DTU sparse subsets) and on different sparsity patterns (clustered vs. uniform camera placements)? Also, how does the method behave with pose noise or imperfect SfM initialization?

---

### Note · Program_Chairs · 2026-01-17
**Submission Desk Rejected by Program Chairs**

The following references in this submission do not refer to real documents and/or have major errors in bibliographic information:

 Zimu Cheng et al. Scgaussian: Structure-consistent gaussian splatting for sparse novel view synthesis. In Neural Information Processing Systems, 2024.